# Towards A Unified Policy Abstraction Theory and Representation Learning Approach in Markov Decision Processes

## Abstract

Lying on the heart of intelligent decision-making systems, how policy is represented and optimized is a fundamental problem. The root challenge in this problem is the large scale and the high complexity of policy space, which exacerbates the difficulty of policy learning especially in real-world scenarios. Towards a desirable surrogate policy space, recently policy representation in a low-dimensional latent space has shown its potential in improving both the evaluation and optimization of policy. The key question involved in these studies is *by what criterion we should abstract the policy space* for desired compression and generalization. However, both the theory on policy abstraction and the methodology on policy representation learning are less studied in the literature. In this work, we make very first efforts to fill up the vacancy. First, we propose a unified policy abstraction theory, containing three types of policy abstraction associated to policy features at different levels. Then, we generalize them to three policy metrics that quantify the distance (i.e., similarity) of policies, for more convenient use in learning policy representation. Further, we propose a policy representation learning approach based on deep metric learning. For the empirical study, we investigate the efficacy of the proposed policy metrics and representations, in characterizing policy difference and conveying policy generalization respectively. Our experiments are conducted in both policy optimization and evaluation problems, containing trust-region policy optimization (TRPO), diversity-guided evolution strategy (DGES) and off-policy evaluation (OPE). Somewhat naturally, the experimental results indicate that there is no a universally optimal abstraction for all downstream learning problems; while the *influence-irrelevance policy abstraction* can be a generally preferred choice.

## 1 Introduction

How to obtain the optimal policy is the ultimate problem in many decision-making systems, such as Game Playing (Mnih et al., 2015), Robotics Manipulation (Smith et al., 2019), Medicine Discovery (Schreck et al., 2019). Policy, the central notion in the aforementioned problem, defines the agent's behavior under specific circumstances. Towards solving the problem, a lot of works carry out studies on policy with different focal points, e.g., how policy can be well represented (Ma et al., 2020; Urain et al., 2020), how to optimize policy (Schulman et al., 2017a; Ho & Ermon, 2016) and how to analyze and understand agents' behaviors (Zheng et al., 2018; Hansen & Ostermeier, 2001).

The root challenge to the studies on policy is the large scale and the high complexity of policy space, especially in real-world scenarios. As a consequence, the difficulty of policy learning is escalated severely. Intuitively and naturally, such issues can be significantly alleviated if we have an ideal surrogate policy space, which are compact in scale while keep the key features of policy space. Related to this direction, low-dimensional latent representation of policy plays an important role in Reinforcement Learning (RL) (Tang et al., 2020), Opponent Modeling (Grover et al., 2018), Fast Adaptation (Raileanu et al., 2020; Sang et al., 2022), Behavioral Characterization (Kanervisto et al., 2020) and etc. In these domains, a few preliminary attempts have been made in devising different policy representations. Most policy representations introduced in prior works resort to encapsulating the information of policy distribution under interest states (Harb et al., 2020; Pacchiano et al., 2020), e.g., learning policy embedding by encoding policy's state-action pairs (or trajectories) and optimizing a *policy recovery* objective (Grover et al., 2018; Raileanu et al., 2020). Rather than policy distribution, some other works resort to the information of policy's influence on the

environment, e.g., state(-action) visitation distribution induced by the policy (Kanervisto et al., 2020; Mutti et al., 2021). Recently, Tang et al. (2020) offers several methods to learn policy representation through policy contrast or recovery from both policy network parameters and interaction experiences. Put shortly, the key question of policy representation learning is *by what criterion we should abstract the policy space* for desired compression and generalization. Unfortunately, both a unified theory on policy abstraction and a systematic methodology on policy representation are currently missing.

In this paper, we make first efforts to fill up the plank in both the theory and methodology. Inspired by the state abstraction theory (Li et al., 2006), first we introduce a unified theory of policy abstraction. We start from proposing three types of policy abstraction: *distribution-irrelevance*, *influence-irrelevance*, and *value-irrelevance*. They follow different abstraction criteria, each of which concerns distinct features of policy. Concretely, we utilize the exact equivalence relations between policies and derive the corresponding policy abstractions. Further, we generalize the exact equivalence relations to policy metrics, allowing quantitatively measure the distance (i.e., similarity) between policies. Such policy metrics are more informative than the binary outcomes of policy equivalence and thus provide more usefulness in policy representation learning. Moreover, towards applying practical policy representation in downstream learning problems, we introduce a policy representation learning approach based on deep metric learning (Kaya & Bilge, 2019). We propose an *alignment loss* for a unified objective function of learning with different policy metrics. The policy representation is learned to render the abstraction criterion through minimizing the difference between the distance of policy embeddings and the quantity measure by the policy metrics. In particular, we use Maximum Mean Discrepancy (Gretton et al., 2012; Nguyen-Tang et al., 2021) for efficient empirical estimation of the policy metrics; and we adopt Layer-wise Permutation-invariant Encoder (Tang et al., 2020) for structure-aware encoding of the parameters of policy network.

In addition to the theoretical understanding of policy abstraction, we further investigate the empirical efficacy of different policy metrics and representations in characterizing policy difference and conveying policy generalization respectively. We conduct experiments in both policy optimization and policy evaluation problems. For policy optimization, we adopt Trust-Region Policy Optimization (TRPO) and Diversity-Guided Evolution Strategy (DGES) as the problem settings from (Kanervisto et al., 2020), covering both gradient-based and gradient-free policy optimization. For policy evaluation, we consider Off-policy Evaluation (OPE). In particular, we establish a series of OPE settings with different configurations of training data and generalization tasks. These settings reflect the circumstances often encountered in RL. Our experimental results indicate that, somewhat naturally, there is no a universally optimal abstraction for all downstream learning problems. Additionally, it turns out that the influence-irrelevance abstraction can be a preferred choice in general cases.

Our main contributions are summarized as follows: 1) We focus on the general policy abstraction problem and to our knowledge, we propose a unified theory of policy abstraction along with several policy metrics for the first time. 2) We propose a unified policy representation learning approach based on deep metric learning. 3) We empirically evaluate the efficacy of our proposed policy representations in multiple fundamental problems (i.e., TRPO, DGES and OPE).

## 2 BACKGROUND

**Reinforcement Learning**  We consider a Markov Decision Process (MDP) (Puterman, 2014) typically defined by a five-tuple $\langle S, A, P, R, \gamma \rangle$, with the state space $S$, the action space $A$, the transition probability $P : S \times A \to \Delta(S)$, the reward function $R : S \times A \to \mathbb{R}$ and the discount factor $\gamma \in [0, 1)$. $\Delta(X)$ denotes the probability distribution over $X$. A stationary policy $\pi : S \to \Delta(A)$ is a mapping from states to action distributions, which defines how to behave under specific states. An agent interacts with the MDP at discrete timesteps by its policy $\pi$, generating trajectories with $s_0 \sim \rho_0(\cdot)$, $a_t \sim \pi(\cdot|s_t)$, $s_{t+1} \sim P(\cdot \mid s_t, a_t)$ and $r_t = R(s_t, a_t)$, where $\rho_0$ is the initial state distribution. We use $P^\pi(s'|s) = \mathbb{E}_{a \sim \pi(\cdot|s)} P(s'|s, a)$ to denote the distribution of next state $s'$ when performing policy $\pi$ at state $s$. For a policy $\pi$, the return $G_t = \sum_{t=0}^{\infty} \gamma^t r_t$ is the random variable for the sum of discounted rewards while following $\pi$, whose distribution is denoted by $Z^\pi$. The value function of policy $\pi$ defines the expected return for state $s$, i.e., $V^\pi(s) = \mathbb{E}_\pi[G_t \mid s_0 = s]$. The goal of an RL agent is to learn an optimal policy $\pi^*$ that maximizes $J(\pi) = \mathbb{E}_{s_0 \sim \rho_0(\cdot)}[V^\pi(s_0)]$.

**Metric Learning**  Here we recall the standard definition of metrics which is central to our work.

**Definition 1** (Metrics (Royden, 1968))**.** *Let $X$ be a non-empty set of data elements and a **metric** is a real-valued function d: $X \times X \to [0, \infty)$ such that for all $x, y, z \in X$: 1) $d(x, y) = 0 \iff x = y$; 2) $d(x, y) = d(y, x)$; 3) $d(x, y) \le d(x, z) + d(z, y)$. A **pseudo-metric** $d$ is a metric with the first condition replaced by $x = y \implies d(x, y) = 0$. The combination $\langle X, d \rangle$ is called a metric space.*

A metric $d$ is often used to quantify the *distance* between two data elements in a general sense. In this paper, we will also use *metric* to stand for *pseudo-metric* for brevity. Typically metric learning aims to reduce the distance between similar data and increase the distance between dissimilar data. With nonlinear transformation offered by deep neural networks, Deep Metric Learning allows us to find such optimal metrics by optimizing a latent representation space of raw data.

## 3 POLICY ABSTRACTION THEORY

Inspired by the state abstraction theory (Li et al., 2006), in this section, we make the first effort in proposing a unified policy abstraction theory. First, we propose the formal definition of three types for policy abstraction; then, we generalize the abstractions to three types of policy metrics. Finally, we analyze the properties of policy abstraction and compare them in several Gridworld MDPs.

### 3.1 POLICY ABSTRACTION

First of all, following the classic definition of an abstraction (Giunchiglia & Walsh, 1992), we propose a general definition of policy abstraction as follows:

**Definition 2** (Policy Abstraction). *A policy abstraction $f : \Pi \to \mathcal{X}$, is a mapping from ground policy space $\Pi$ to an abstract space $\mathcal{X}$. $f(\pi) \in \mathcal{X}$ is the abstract policy representation corresponding to ground policy $\pi \in \Pi$, and the inverse image $f^{-1}(\chi)$ with $\chi \in \mathcal{X}$, is the set of ground policies that correspond to $\chi$ under abstraction function $f$.*

It is apparent that there are many such abstractions since we may have many possible ways to partition the policy space. However, we are only interested in some useful ones among them that follow specific abstraction criteria to preserve the important features related to decision making. In this paper, we present three types of policy abstraction which are defined below:

**Definition 3.** *Given an MDP and a ground policy space $\Pi$, for any two policies $\pi_i, \pi_j \in \Pi$, we define three types of policy abstraction as follows:*

1. *A distribution-irrelevance abstraction ($f_\pi$) is such that for all $s \in S$, $a \in A$, $f_\pi(\pi_i) = f_\pi(\pi_j)$ implies that $\pi_i(a \mid s) = \pi_j(a \mid s)$.*
2. *An influence-irrelevance abstraction ($f_{P^\pi}$) is such that for all $s, s' \in S$, $f_{P^\pi}(\pi_i) = f_{P^\pi}(\pi_j)$ implies that $P^{\pi_i}(s'|s) = P^{\pi_j}(s'|s)$.*
3. *A value-irrelevance abstraction ($f_{V^\pi}$) is such that for all $s \in S$, $f_{V^\pi}(\pi_i) = f_{V^\pi}(\pi_j)$ implies that $V^{\pi_i}(s) = V^{\pi_j}(s)$.*

These abstractions aggregate policies based on the corresponding equivalence relations with respective concerns on different features of policy. Intuitively, the distribution-irrelevance abstraction ($f_\pi$) preserves the action distribution of the policy; the influence-irrelevance abstraction ($f_{P^\pi}$) preserves the state transition distribution induced by the policy, i.e., the influence caused by the policy on the environment; and value-irrelevance abstraction ($f_{V^\pi}$) preserves the value function of the policy. In addition to the policy abstractions introduced in Definition 3, we provide some other ones in Appendix A.2. Moreover, we revisit the policy abstractions adopted in prior related works and summarize them from the angle of our policy abstraction theory in Table 4 of Appendix B.

### 3.2 POLICY METRICS

The policy abstractions allow us to aggregate policies according to equivalence relation. However, exact equivalence is rarely encountered in continuous policy space (e.g., the usual case with neural policies), thus useful abstraction can be seldom obtained. Moreover, the equivalence relation offers only qualitative (i.e., binary) outcomes and is incapable of measuring the similarity between policies, which is significant to policy representation learning. To this end, we generalize the policy abstractions to policy metrics which quantitatively measures the distance between two policies.

Corresponding to the three types of policy abstraction, we define the following three policy metrics:

**Definition 4.** *Given an MDP, a ground policy space $\Pi$, a state distribution $p(s)$ and a distribution (pseudo-)metric $D(\cdot, \cdot)$, for any two policies $\pi_i, \pi_j \in \Pi$, we define three policy metrics as follows:*

1. *A distribution-irrelevance metric: $d_\pi(\pi_i, \pi_j) = \mathbb{E}_{s \sim p(s)}[D(\pi_i(a \mid s), \pi_j(a \mid s))]$.*
2. *An influence-irrelevance metric: $d_{P^\pi}(\pi_i, \pi_j) = \mathbb{E}_{s \sim p(s)}[D(P^{\pi_i}(s' \mid s), P^{\pi_j}(s' \mid s))]$.*
3. *A value-irrelevance metric: $d_{V^\pi}(\pi_i, \pi_j) = \mathbb{E}_{s \sim p(s)}[D(Z^{\pi_i}(s), Z^{\pi_j}(s))]$.*

Table 1: Properties of different policy abstraction.

| Abstraction | Abstraction Criterion (for $\pi_1, \pi_2, \forall s, s', a \in S^2 \times A$) | Fineness | Task Relevance |
|:---:|:---:|:---:|:---:|
| $f_\Theta$ | Policy Parameter Equivalence ($\theta_1 = \theta_2$) | Highest | None |
| $f_\pi$ | Action Distribution Equivalence ($\pi_i(a \mid s) = \pi_j(a \mid s)$) | High | Low |
| $f_{P^\pi}$ | Dynamics Influence Equivalence ($P^{\pi_i}(s'|s) = P^{\pi_j}(s'|s)$) | Middle | Middle |
| $f_{V^\pi}$ | Value Function Equivalence ($V^{\pi_i}(s) = V^{\pi_j}(s)$) | Low | High |
| $f_0$ | Triviality (taking all policies as the same) | Lowest | None |

These metrics follow the same abstraction criteria as in Definition 3, i.e., the irrelevance regarding action distribution, influence and value, measuring the similarity of policies by the distance at respective levels. Compared to the binary outcomes offered by the equivalence relations, the metrics defined here are continuous, thus are more informative in comparing and representing policies in finer views. Specially, one may see that the equivalence relations used in Definition 3 induce corresponding discrete pseudo-metrics, e.g., $d_\pi^{\text{Eq}}(\pi_i, \pi_j) = 0$ if $f_\pi(\pi_i) = f_\pi(\pi_j)$, and 1 otherwise. Notice the metrics proposed above depends on the distribution metric $D$ and state distribution $p(s)$. For $D$, typical choices can be Jeffreys Divergence (Jeffreys, 1946) and Maximum Mean Discrepancy (MMD) (Nguyen-Tang et al., 2021). For $p(s)$, intuitively, it should be the distribution of states we are interested in when comparing two policies. We defer the concrete choices for practical implementation of these metrics in Section 4.

### 3.3 PROPERTIES OF THE ABSTRACTIONS

Superficially, the three abstractions proposed preserve features that are progressively more relevant to decision making in the learning task, but essentially, what is the relationship between the three abstractions? To investigate the problem, we define the *fineness* of policy abstractions similar to the one for state abstractions used in (Li et al., 2006), to prove how the three abstractions are related.

**Definition 5** (Abstraction Fineness). *Let $F_\Pi$ denotes the set of abstractions on ground policy space $\Pi$. Suppose $f_1, f_2 \in F_\Pi$. We say $f_1$ is finer than $f_2$, denoted $f_1 \succeq f_2$, iff $\forall \pi_1, \pi_2 \in \Pi$, $f_1(\pi_1) = f_1(\pi_2)$ implies $f_2(\pi_1) = f_2(\pi_2)$. If, $f_1 \neq f_2$, then $f_1$ is strictly finer than $f_2$, denoted $f_1 \succ f_2$. In contrast, we may also say $f_2$ is (strictly) coarser than $f_1$, denoted $f_2 \preceq f_1$ ($f_2 \prec f_1$).*

It is easy to see the relation $\succeq$ satisfies self-reflexivity, antisymmetry and transitivity, thus it is a partial ordering. Consider the set of possible policy abstractions, while the coarsest abstraction ($f_0$) is the trivial representation where all policies are treated as the same; while the finest abstraction is the identity representation, e.g., $f_\Theta(\pi_\theta) = \theta$ for a policy neural network parameterized with $\theta \in \Theta$. With the partial ordering $\succeq$, we further derive the following theory.

**Theorem 3.1.** *Under the Definition 3 and 5, if the reward function $R$ depends only on state $s \in S$, we have $(f_\Theta \succeq) f_\pi \succeq f_{P^\pi} \succeq f_{V^\pi} (\succeq f_0)$.*

The proof is provided in Appendix A.1 along with more discussion on other cases of the reward function. The theorem declares how the three policy abstractions are related to each other in the sense of abstraction fineness with the two extreme cases ($f_\Theta, f_0$) for reference. The coarser the abstraction is, the more the original policy space is abstracted.

In Table 1, we summarize the properties of different policy abstractions, regarding abstraction criteria, abstraction fineness and task relevance. The major conclusion is that there is an inverse relation between abstraction fineness and task relevance. Except for the two extreme cases ($f_\Theta, f_0$) that are totally task-independent, the policy abstraction becomes more task-relevant as the abstraction criterion concerns more policy features related to the learning task. The distribution-irrelevance abstraction $f_\pi$ concerns the policy behavior in the learning task, defined by action distribution at interested states; meanwhile $f_\pi$ is coarser than $f_\Theta$ since the same policy behavior can be realized by non-unique policy parameter. Taking one step closer to the task, the influence-irrelevance abstraction $f_{P^\pi}$ cares about the state transition dynamics induced by policy behavior. Obviously, $f_{P^\pi}$ is coarser than $f_\pi$ as different behaviors may induce the same transition distribution. The value-irrelevance abstraction $f_{V^\pi}$ further involves the rewards of long-term dynamics, thus is the most task-relevant and coarsest among the three types of policy abstraction.

**Empirical Comparison of Policy Metrics in Gridworld MDPs** To compare these policy abstractions in a quantitative view, we demonstrate how the distances of two policies measured by the corresponding policy metrics differ in several Gridworld MDPs. We use *Distinct Policies*, *Doorway* from (Kanervisto et al., 2020) and design a new task, *Key Action* for simple prototypes of tasks with

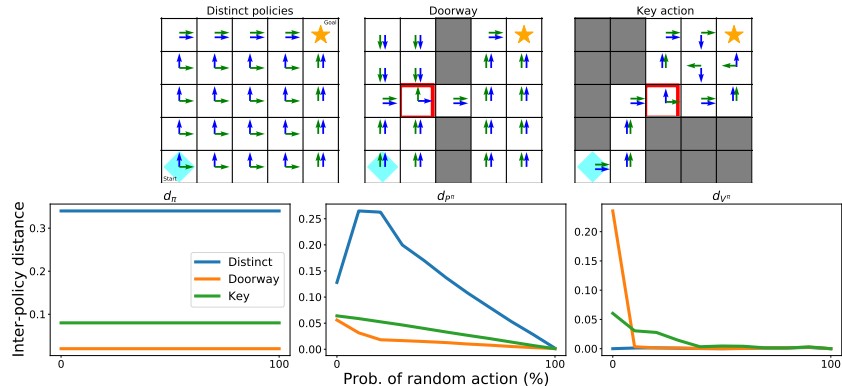

Figure 1: Policy comparison with different policy metrics in Gridworld. *Top Panel:* The illustration of three Gridworld MDPs and two deterministic policies (blue and green). *Bottom Panel:* The distance curves of the two policies measured by $d_\pi, d_{P^\pi}, d_{V^\pi}$ ($y$-axi), against the stochasticity of the environment ($x$-axi). $d_{P^\pi}$ is able to distinguish the two policies across all the settings.

different features; moreover, we increase the stochasticity of the environment for a better evaluation. In particular, $\mathbb{E}_{s\sim p(s)}D(\cdot,\cdot)$ is calculated by average the absolute differences over all states. The illustrations and results are shown in Fig. 1 while more results can be found in Appendix C.

We observe that the distribution-irrelevance metric $d_\pi$ may fail to show the difference in dynamics and outcome between the two policies, e.g., in Doorway. This is because $d_\pi$ measures the difference in the action distribution itself, i.e., independent of the dynamics as well as the increasing stochasticity of the environment. This issue may be resolved by using a designated distribution that concentrates over key states. Conversely, the value-irrelevance $d_{V^\pi}$ measures the difference in the outcomes of the two policies regardless their differences in action distribution and dynamics, e.g., in Distinct Policies. Another discovery is that $d_{V^\pi}$ quickly degenerates and turns to be not informative as the increase of stochasticity, showing its poor robustness. By contrast, the influence-irrelevance metric $d_{P^\pi}$ is a *sweet* intermediate point, consistently keeping the ability of distinguishing the two policies across all the environments and stochasticity configurations. In a summary, different policy abstractions and metrics may yield different outcomes for the same two policies and the optimality depends on the specific downstream learning problem concerned. Later, we evaluate these options in representative downstream RL problems in Section 5 and 6 for useful insights.

## 4 POLICY REPRESENTATION LEARNING APPROACH

The next question concerned in practice is: *how can we learn the representation of RL policies (usually modeled by NNs) in a general way?* Based on the policy metrics introduced above, we propose a policy representation learning approach by following the principle of Deep Metric Learning.

### 4.1 LEARNING POLICY REPRESENTATION BY EMBEDDING ALIGNMENT

The policy metrics proposed in previous section measure the quantitative relationship between policies from different perspectives of the policy abstraction criteria. For a unified objective function of learning from different policy metrics, we use the *alignment loss*, with which the difference between the distances of two policies in the representation space and in the policy metric space is minimized. Concretely, consider a policy representation function $f_\psi$, and the alignment loss can be formalized as,

$$\mathcal{L}_{\mathrm{AL}}(\psi) = \mathbb{E}_{\pi,\pi'\in\Pi}\left[\left(\|f_\psi(\pi) - f_\psi(\pi')\|_2 - \eta d_*\left(\pi,\pi'\right)\right)^2\right], \tag{1}$$

where we consider $d_* \in \{d_\pi, d_{P^\pi}, d_{V^\pi}\}$ and $\eta$ is the weight for scaling. Similar forms of the alignment loss are also adopted in the studies on state representation learning (Zhang et al., 2020).

As we can see, $\mathcal{L}_{\mathrm{AL}}(\psi)$ consists of two metrics, i.e., the $L_2$ distance function ($\|\cdot\|_2$) of two inputs and the policy metric ($d_*(\cdot,\cdot)$). Intuitively, minimizing the alignment loss is to align the two metrics by optimizing the policy representation function $f_\psi$. By this means, we are able to learn different policy representation functions, which maps the ground policy $\pi \in \Pi$ to the latent embedding $\chi_\pi = f_\psi(\pi) \in \mathcal{X}$. The embedding preserves the policy features corresponding to the abstraction criteria reflected by the specific policy metric considered. For a practical implementation, the following problems are the estimation of policy metrics $d_*$ and the realization of the training for policy representation function $f_\psi$, which are detailed in the next two subsections respectively.

In the literature of learning policy representation, representative methods follow the principle of behavior recovering (Grover et al., 2018) and policy contrast (Tang et al., 2020). To our knowledge, none of prior works take a systematic view of policy abstraction. In Table 4, we show that prior methods are specific instances of one of our proposed policy abstractions that differ in realization.

## 4.2 ESTIMATING POLICY METRICS VIA MAXIMUM MEAN DISCREPANCY

Given a tractable metric $D$ and a state distribution $p$, the policy metrics (i.e., $d_\pi$, $d_{P^\pi}$, $d_{V^\pi}$) can be calculated exactly if the probability distributions (i.e., $\pi$, $P^\pi$, $Z^\pi$) are available. However, this is usually infeasible in practice; instead, in more regular cases, only finite samples of policy interaction are available. Although the empirical distributions can be estimated in simple MDPs where the state-action space is finite (as in Appendix D), unfortunately, approximating the distributions and computing the metrics are non-trivial, especially with high-dimensional continuous state-action space.

Therefore, we estimate the policy metrics directly from the samples, bypassing estimating the empirical distributions (i.e., $\widetilde{\pi}$, $\widetilde{P}^\pi$, $\widetilde{Z}^\pi$). In particular, we adopt MMD (Gretton et al., 2012; Nguyen-Tang et al., 2021) as the distribution metric, i.e., let $D$ be $D_{\text{MMD}}$. MMD measures the maximum value of the mean discrepancy of two distributions regarding all possible functions in a predefined family. Conventionally, let the class of functions $h : X \to \mathbb{R}$ be a unit ball in a Reproducing Kernel Hilbert Space (RKHS) $\mathcal{H}$ associated with a continuous kernel $k(\cdot, \cdot)$ on $X$, $p, q$ be two distribution defined on $X$, $x, x'$ and $y, y'$ be i.i.d. samples from $p$ and $q$ respectively, the MMD is defined as:

$$D_{\text{MMD}}(p, q; \mathcal{H}) = \sup_{h \in \mathcal{H}: \|h\|_{\mathcal{H}} \le 1} (\mathbb{E}_{x \sim p}[h(x)] - \mathbb{E}_{y \sim q}[h(y)]) = \|\mu_p - \mu_q\|_{\mathcal{H}}$$

$$= \left( \mathbb{E}_{x,x'}[k(x, x')] + \mathbb{E}_{y,y'}[k(y, y')] - 2\mathbb{E}_{x,y}[k(x, y)] \right)^{\frac{1}{2}}, \quad (2)$$

where $\mu_p = \int_X k(x, \cdot) p(\mathrm{d}x)$ is the mean embedding of $p$ into $\mathcal{H}$(Smola et al., 2007). Thus, MMD can be empirically estimated with samples $\{x_i\}_{i=1}^N \sim p$ and $\{y_i\}_{i=1}^M \sim q$:

$$\widetilde{D}_{\text{MMD}}^2(\{x_i\}, \{y_i\}; k) = \frac{1}{N^2} \sum_{i,j} k(x_i, x_j) + \frac{1}{M^2} \sum_{i,j} k(y_i, y_j) - \frac{2}{NM} \sum_{i,j} k(x_i, y_j). \quad (3)$$

According to Eq. 3, we can estimate the policy metrics $d_\pi, d_{P^\pi}, d_{V^\pi}$ empirically from the samples $\{a_i\}, \{s_i'\}, \{G_i\}$ of different policies respectively, under the sampled states $\{s_i\}$ for the expectation $\mathbb{E}_{p(s)}$. However, it is often impractical to obtain multiple samples under the same state. Thus, we resort to estimating the surrogates, e.g., $\hat{d}_{P^\pi}(\pi_i, \pi_j) = D(P^{\pi_i}(s, s'), P^{\pi_j}(s, s'))$ for $d_{P^\pi}$, where the joint distributions rather than the state-conditioned distributions are measured. We use Gaussian kernel by default, i.e., $k(x, x') = \exp\left(-\frac{\|x - x'\|_2^2}{2\sigma^2}\right)$. Consequently, the empirical estimates of the policy metrics serve as the self-supervision in Eq. 1.

## 4.3 REALIZING THE TRAINING OF POLICY REPRESENTATION FUNCTION

With the empirical policy metrics provided in previous section, the training of policy representation is straightforward with a differentiable function $f_\psi$ by optimizing the alignment loss (Eq. 1). The realization of policy representation function concerns two aspects: 1) the choice of policy data (or original representation) and 2) the construction of $f_\psi$ (i.e., how policy data is encoded).

For the first aspect, we focus on parameterized policy $\pi_\theta$ (typically by a neural network) and use policy parameter $\theta$ as the policy data. One may recall that $\theta$ itself can be viewed as the finest representation obtained by policy abstraction $f_\Theta$ in Table 1. Such an original representation (i.e., $\theta$) is high-dimensional and highly nonlinear, offering no help in the compression and generalization of policy space. In addition, we are aware that in some cases the policy parameters may be not available, and thus the interaction experiences generated by the policy can be alternative policy data, as used in (Grover et al., 2018; Tang et al., 2020). Our policy representation learning approach is compatible with such alternatives with the need of possible slight modifications. For the second aspect, we adopt Layer-wise Permutation-invariant Encoder (LPE) (Tang et al., 2020) as the implementation choice of $f_\psi$ (see Fig. 5), which has demonstrated the effectiveness in encoding conventional policy networks. To be specific, for the parameter $\theta = \{W_i, b_i\}_{i=0}^k$ of policy $\pi$, i.e., the weights and biases of $k$-layer MLP,[1] the weight $W_i \in \mathbb{R}^{l_i \times l_{i+1}}$ and bias $b_i \in \mathbb{R}^{1 \times l_{i+1}}$ ($l_i$ is the unit number of the $i$-layer; $l_0$ and

---

[1]The activation function is not considered since the structure is fixed for policies in convention RL setting. In principle, LPE can be generalized to tailor other advanced network structure.

$l_k$ are for the input and output layers) are concatenated ($\oplus$) and transposed, followed by a MLP ($f_{\psi,i}$) and a mean-reduce operation (MR), resulting in a layer embedding $z_i$; Thereafter, the policy embedding is obtained by concatenating the embedding of each layer. Formally,

$$z_i = \text{MR}\left(f_{\psi,i}\left([W_i \oplus b_i]^\top\right)\right) = \frac{1}{l_i+1}\sum_{j=1}^{l_i+1} f_{\psi,i}\left(([W_i \oplus b_i]^\top)_{j,\cdot}\right), \quad \chi_{\pi_\theta} = f_\psi(\theta) = \bigoplus_{i=0}^{k} z_i \quad (4)$$

Each row of $[W_i \oplus b_i]^\top$, indexing by the subscript $j, \cdot$, describes a transformation of the $i$-layer into the next layer. All the rows are fed into $f_{\psi,i}$ separately and are then averaged into $z_i$. In a consequence, the policy embedding serves as the compact representation of the policy network by summarizing the transformations made by the each layer of it. The significant difference between LPE and a straightforward MLP encoder is that, LPE provides structure-aware representation, i.e., both the intra-layer and inter-layer structures are explicitly considered. Intuitively, this alleviates the difficulty of learning representation from the policy network parameters. Other advanced encoder structures are beyond the scope of this work and we leave them as future work.

Till now, we can update the parameters of LPE $\psi = \{\psi_i\}_{i=0}^{k}$ by optimizing Eq. 1 with the policy samples from some given set of policies and the empirical policy metrics estimated accordingly. Depending on the specific choice of policy metric, the policy representation is learned to render the policy abstraction in Table 1, starting from $f_\Theta$ and going downwards to the corresponding level.

## 5 APPLYING POLICY ABSTRACTION TO POLICY OPTIMIZATION

Despite the theoretical understanding of the policy abstraction, we have no idea about how the derived policy metrics behave in different downstream learning problems. To shed some light on this, we evaluate the efficacy of the policy metrics proposed in Sec. 3.2 in policy optimization, including Trust-Region Policy Optimization (TRPO) and Diversity-Guided Evolutionary Strategy (DGES).

**Trust-Region Policy Optimization** We adopt TRPO problem as the first test stone for our policy abstractions. Specifically, the objective of TRPO problem is to maximize the policy return while constraining the difference between old and new policies: $J_{\text{TRPO}}(\theta) = \mathbb{E}_{\tau \sim \mathbb{P}_{\pi_\theta}}[\mathcal{R}(\tau)]$, $s.t., d_*(\pi_\theta, \pi_{\theta_{old}}) \leq \sigma$, where $\sigma$ is a threshold. For our experiments, we consider the policy metrics $d_* \in \{d_\pi, d_{P^\pi}, d_{V^\pi}\}$. In another word, the learning agent checks if the difference measured by the policy metrics are larger than $\sigma$ for each policy update. In this experiment, the original TRPO (Schulman et al., 2015) is generalized to incorporate different alternative metrics for the trust-region constraint. Thus, we can evaluate the efficacy of the different trust regions provided by our proposed policy metrics, shedding some light on what policy features we care the most in TRPO.

We adopt a Gridworld environment where the agent can move to one of $N$ directions at each grid and only one direction yields high reward (Kanervisto et al., 2020). The results are shown in Fig. 2(a). We observe that all our TRPO variants (i.e., TRPO-$f_*$) outperform Vanilla-PO (i.e., no trust-region constraint used), demonstrating the effectiveness of our policy abstractions. Moreover, TRPO-$f_\pi$ outperforms the others. This is because $f_\pi$ follows the abstraction criterion regarding action distribution, thus pertains to the essence

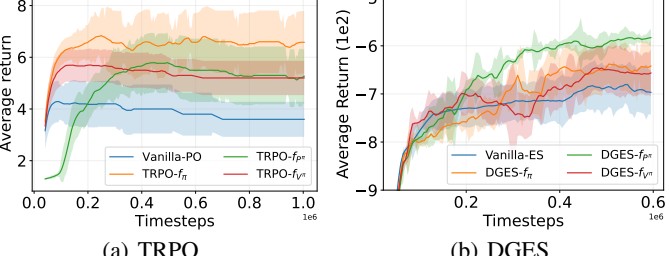

(a) TRPO  (b) DGES

Figure 2: Performance of different policy abstractions in: *(a)* Trust-Region Policy Optimization (TRPO) for a Gridworld environment; and *(b)* Diversity-Guided Evolution Strategy (DGES) for a Point environment. Results are the mean and half a standard deviation (shaded) over 10 and 5 trials for TRPO and DGES respectively.

of TRPO. By contrast, $f_{P^\pi}$ and $f_{V^\pi}$ utilize coarser abstraction which does not hold the features of action distribution. In addition, we demonstrate the superiority of our policy abstractions when compared with existing related methods in Appendix E.1.

**Diversity-Guided Evolution Strategy** Next, we adopt DGES problem as the second test stone for our policy abstractions. Formally, the objective of DGES problem is to maximize the policy return of the current policy $\pi_\theta$ and maximize its policy difference to the ancestor policy $\bar{\pi}$: $J_{\text{DGES}}(\theta) = \mathbb{E}_{\tau \sim \mathbb{P}_{\pi_\theta}}[\mathcal{R}(\tau)] + \beta \sum_{p=1}^{N} d_*(\pi_\theta, \bar{\pi})$, where $\beta \geq 0$ is the weight. Similarly, we consider the policy

metrics $d_* \in \{d_\pi, d_{P^\pi}, d_{V^\pi}\}$. Here, the choices of policy metrics realize the population diversity in different ways. We aim at exploring the diversity concerning which policy feature is the most effective in DGES.

To explore this, we leverage the Point environment with deceptive rewards (Pacchiano et al., 2020). The results are reported in Fig. 2(b). In comparison to Vanilla-ES (i.e., $\beta = 0$), optimizing policy diversity (i.e., $\beta > 0$) based on our policy metrics (i.e., DGES-$f_*$) does help exploration and thus leads to better performance. In particular, DGES-$f_{P^\pi}$ performs the best. Since ES optimizes policy in a gradient-free fashion, the evolution process concerns only policy return. Therefore, the distribution-irrelevance abstraction $f_\pi$ (i.e., the winner in the TRPO experiment) can be redundant since multiple action distributions may have the same outcome (i.e., influence and value). For the value-irrelevance abstraction $f_{V^\pi}$, it turns to be too fine to contain the features of policy behavior (i.e., action distribution and influence). Therefore, the influence-irrelevance abstraction $f_{P^\pi}$ serves as a sweet point. Furthermore, we provide additional comparative evaluation in Appendix E.2.

## 6 APPLYING POLICY ABSTRACTION TO OFF-POLICY EVALUATION

After the investigation in policy optimization, now we move to policy evaluation. Typical OPE (Fu et al., 2021; Harb et al., 2020) focuses on using offline data to evaluate unseen policies. Likewise, we are interested in studying the value generalization performance on unseen policies of the representations learned regarding different policy abstractions. The appealing characteristic of policy representation in value generalization has been studied in (Tang et al., 2020), where a Policy-extended Value Function Approximator (PeVFA, $\mathbb{V}(\chi_\pi)$) takes as input the policy representation $\chi_\pi$ approximates the values of multiple policies and offers implicit value generalization among the policy representation space.

For policy data collection, we run PPO (Schulman et al., 2017b) in OpenAI Gym continuous control tasks: InvertedDoublePendulum-v2 (IDP-v2) and LunarLanderContinuous-v2 (LLC-v2) (Brockman et al., 2016). By collecting the policies at intervals during the learning process, we build an offline policy set, based on which we train our policy representations and a PeVFA $\mathbb{V}(\chi_\pi)$. For concrete problem settings, we establish both weak and strong generalization OPE scenarios which differs at the difficulty of evaluating the unseen policies. For the weak generalization scenario (*easy*), we sample training data uniformly from the whole band of the policy set. For the strong generalization scenario (*hard*), we separate the policy set and use the low-performance policies for the training data, with the rest taken as the unseen policies to evaluate. For both the settings, the ratio of sampling and separation is set to be 20%, 40%, or 80%. We report the results of the ratio 20% in Table 2 and leave the results of other ratios in Appendix G. For evaluation protocols, we report the evaluation (testing) error of unseen policies (**T-error**) and the generalization gap (**G-gap**), i.e., the difference between training and testing error. We denote different policy representations by their underlying policy abstraction (e.g., $f_\pi$) correspondingly. Complete details can be found in Appendix F.

**Weak Generalization Scenario in OPE**  First, we study the empirical comparison in the weak generalization scenario. Table 2 reports the results of value generalization for the policy representations learned based on corresponding policy abstractions. To be specific, the $f_\Theta$ denotes directly using policy parameters $\theta$ as policy representations (i.e., no representation training). For our proposed policy abstractions $f_\pi, f_{P^\pi}, f_{V^\pi}$, we learn the representations for them according to Eq. 1 based on the LPE and MMD estimation (Sec. 4.3). To further complete the comparison, we include two additional representations $f_{RE}$ and $f_{EL}$: $f_{RE}$ uses a randomly initialized LPE with no further training while $f_{EL}$ uses the LPE trained by the end-to-end OPE loss (see Appendix F.2) respectively. Note that $f_{EL}$ can be viewed as a variant of $f_{V^\pi}$ since it also learns from values but does not optimize the alignment loss. Besides, we also include the representation ($f_{CL}$) learned by unsupervised contrastive learning based on InfoNCE loss (van den Oord et al., 2018), as proposed in (Tang et al., 2020).

From the Table 2 (Weak Generalization), we can observe that $f_\pi, f_{P^\pi}, f_{V^\pi}$ outperforms $f_\Theta, f_{RE}$, and $f_{EL}$ in both IDP-v2 and LLC-v2. This demonstrates the effectiveness and superiority of our proposed representations in value function approximation and generalization. $f_{EL}$ is significantly better than the $f_\Theta$ and $f_{RE}$, indicating the advantages of LPE structure and training. We can observe that contrastive policy representation $f_{CL}$ performs poorly. We postulate that with less training data available at the 20% sampling ratio, the $f_{CL}$ with emphasis on policy instance-level comparison suffers from higher evaluation error and generalization gap. The superiority of $f_{V^\pi}$ compared to $f_{EL}$ from the Table 2 demonstrates the effectiveness of alignment loss. This is because although both $f_{V^\pi}, f_{EL}$ learn policy representation from the information of policy value, naive end-to-end training is less effective than alignment optimization which establishes the representation space based on the

Table 2: Performance of different policy abstractions in Off-policy Evaluation (OPE). The minimum value for each task is highlighted. Results are the mean $\pm$ a standard deviation over 10 and 5 trials (for *weak* and *strong* respectively). The $f_{V\pi}$ has lower T-error and G-gap on both the generalization tasks.

| Env | Abstraction | Weak Generalization | | Strong Generalization | |
|---|---|---|---|---|---|
| | | T-error | G-gap | T-error | G-gap |
| IDP-v2 | $f_\Theta$ | $0.0059 \pm 0.0008$ | $0.0039 \pm 0.0006$ | $0.1592 \pm 0.0107$ | **$0.0778 \pm 0.0437$** |
| | $f_{RE}$ | $0.0056 \pm 0.0009$ | $0.0038 \pm 0.0010$ | $0.1676 \pm 0.0086$ | $0.1674 \pm 0.0087$ |
| | $f_{EL}$ | $0.0048 \pm 0.0003$ | $0.0027 \pm 0.0008$ | $0.1783 \pm 0.0060$ | $0.1712 \pm 0.0145$ |
| | $f_{CL}$ | $0.0067 \pm 0.0010$ | $0.0046 \pm 0.0008$ | $0.1567 \pm 0.0081$ | $0.1491 \pm 0.0107$ |
| | $f_\pi$ | $0.0044 \pm 0.0003$ | $0.0025 \pm 0.0006$ | $0.1812 \pm 0.0013$ | $0.1803 \pm 0.0011$ |
| | $f_{P\pi}$ | **$0.0044 \pm 0.0003$** | $0.0024 \pm 0.0006$ | $0.1789 \pm 0.0045$ | $0.1778 \pm 0.0049$ |
| | $f_{V\pi}$ | $0.0046 \pm 0.0003$ | **$0.0022 \pm 0.0005$** | **$0.1320 \pm 0.0093$** | $0.1295 \pm 0.0114$ |
| LLC-v2 | $f_\Theta$ | $0.0018 \pm 0.0005$ | $0.0016 \pm 0.0003$ | $0.1898 \pm 0.0237$ | $0.0926 \pm 0.1592$ |
| | $f_{RE}$ | $0.0028 \pm 0.0007$ | $0.0025 \pm 0.0007$ | $0.0729 \pm 0.0197$ | $0.0718 \pm 0.0196$ |
| | $f_{EL}$ | $0.0017 \pm 0.0004$ | $0.0016 \pm 0.0004$ | $0.0656 \pm 0.0088$ | $0.0646 \pm 0.0092$ |
| | $f_{CL}$ | $0.0035 \pm 0.0005$ | $0.0032 \pm 0.0004$ | $0.0589 \pm 0.0176$ | $0.0572 \pm 0.0188$ |
| | $f_\pi$ | $0.0015 \pm 0.0005$ | $0.0013 \pm 0.0005$ | $0.1365 \pm 0.0367$ | $0.1318 \pm 0.0332$ |
| | $f_{P\pi}$ | $0.0015 \pm 0.0004$ | $0.0013 \pm 0.0004$ | $0.0905 \pm 0.0402$ | $0.0900 \pm 0.0404$ |
| | $f_{V\pi}$ | **$0.0014 \pm 0.0003$** | **$0.0011 \pm 0.0003$** | **$0.0473 \pm 0.0043$** | **$0.0470 \pm 0.0042$** |

policy metrics. In general, the value generalization results among our abstractions $f_\pi, f_{P\pi}, f_{V\pi}$ do not differ much. This is mainly because in the weak generalization setting, the unseen policies obey the same distribution as the training policies, thus posing less difficulty of value generalization.

**Strong Generalization Scenario in OPE**   Now we move to the study in the strong generalization scenario and similarly the results are reported in Table 2 (Strong Generalization). Compared to the weak generalization scenario, the overall T-error and G-gap are significantly higher in the strong generalization scenario. This is reasonable because there is a larger performance difference between the training and unseen policies. In other words, the unseen policies belong to out-of-distribution data. The $f_{V\pi}$ obtains the lowest evaluation error on the two environments, which indicates the value-irrelevance abstraction with higher task relevance may be best suited for the strong generalization setting. For the explanation, since the objective of OPE lies at the value function approximation and generalization, we consider that the value-irrelevance principle of $f_{V\pi}$ is consistent to the objective and thus fits naturally.

With only low-performance policies for the training data at the 20% sampling ratio (*hardest*), the results of other abstractions including our proposed $f_\pi$ and $f_{P\pi}$ on the task are poor. The main reason may be that under the strong generalization setting, there is a large data-shift between the training and unseen policies. $f_\pi$ and $f_{P\pi}$ fails to learn a policy abstraction with generalization ability in the absence of diversity policies. Nevertheless, from the Table9, 10, as the sampling ratio increase and training policies become more diverse, the advantage of $f_\pi$ and $f_{P\pi}$ over other policy abstractions gradually emerge. Moreover, in the hardest case, $f_{P\pi}$ is better than $f_\pi$, which shows that the influence-irrelevance policy abstraction may be a general policy abstraction option.

For the other baselines, $f_\Theta$ still shows few competition. For $f_{RE}, f_{EL}, f_{CL}$, they falls behind $f_{V\pi}$ while slightly outperforms $f_{P\pi}$ and $f_\pi$ in Table 2. Such slight advantages no long holds in the settings of higher sampling ratios (i.e., 40% in Table 9 and 80% in Table 10). Unlike weak generalization scenario, $f_{CL}$ is not so bad in strong generalization scenario. The main reason is that encountering hard policy evaluation tasks (Strong Generalization), other methods suffer from performance degradation and are no longer superior to contrastive learning. In contrast, contrastive learning based on policy instance-level comparison maintains a relatively good result.

**Other Experiments**   In addition to Table 2, we provide more results under different settings of data amount in Appendix G.1, for both the weak and strong generalization scenarios. Other comprehensive studies (e.g., extrapolation behaviors, visualization) can be found in Appendix G.2,G.3.

## 7   CONCLUSION & LIMITATIONS

In this work, we introduce a unified policy abstraction theory, including three major types of policy abstraction, and corresponding policy metrics derived from the abstraction, as well as the analysis of their properties. We further propose a policy representation learning approach based on deep metric learning. We empirically evaluate the efficacy of different policy abstraction in both policy optimization (i.e., TRPO, DGES) and off-policy evaluation (OPE). For limitations and future work, we only provide the theory on the fineness of policy abstraction, while provide no theory on the optimality, although the optimality ought to depend on the downstream problem considered. For policy representation learning, the alignment loss and MMD metric are not the only choices; besides, other representation learning principles (Bardes et al., 2021) are potential.

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

# A  MORE CONTENT ON POLICY ABSTRACTION THEORY

## A.1  PROOF OF THEOREM 3.1

*Proof.* We prove the partial ordering ($\succeq$) of the Theorem 3.1 one by one in the following.

❶ $f_\pi \succeq f_{P^\pi}$. Given an MDP $M$, two policies $\pi_i, \pi_j \in \Pi$. We define $P$ as the transition probability and use $P^\pi(s'|s) = \mathbb{E}_{a \sim \pi(\cdot|s)} P(s'|s, a)$ to denote the distribution of next state $s'$ when performing policy $\pi$ at state $s$, respectively. Then, we have

$$
\begin{aligned}
P^{\pi_i}(s'|s) = \mathbb{E}_{a \sim \pi_i(\cdot|s)} P(s'|s, a), \\
P^{\pi_j}(s'|s) = \mathbb{E}_{a \sim \pi_j(\cdot|s)} P(s'|s, a).
\end{aligned}
\tag{5}
$$

If $f_\pi(\pi_i) = f_\pi(\pi_j)$, with the Definition 3, we have $\forall s \in S, \forall a \in A, \pi_i(a|s) = \pi_j(a|s)$. Combined with Eq. 5, we derive,

$$
\forall s, s' \in S, P^{\pi_i}(s'|s) = P^{\pi_j}(s'|s).
\tag{6}
$$

Recall the definition of $f_{P^\pi}$, we can obtain $f_\pi \succeq f_{P^\pi}$.

❷ $f_{P^\pi} \succeq f_{V^\pi}$. Given an MDP $M$, two policies $\pi_i, \pi_j \in \Pi$, a reward function $R$. Start with the definition of the value function $V^\pi(\cdot)$, we consider the two cases:
**Case 1**. The reward function $R$ depends only on state $s \in S$, we derive the value function:

$$
\forall s \in S, V^\pi(s) = R(s) + \sum_{s'} P^\pi(s' \mid s) V^\pi(s').
\tag{7}
$$

If $f_{P^\pi}(\pi_i) = f_{P^\pi}(\pi_j)$, with the Definition 3, we have $\forall s, s' \in S, P^{\pi_i}(s'|s) = P^{\pi_j}(s'|s)$. When the Eq. 7 holds, we have:

$$
\forall s \in S, V^{\pi_i}(s) = V^{\pi_j}(s).
\tag{8}
$$

Recall the definition of $f_{V^\pi}$, we can obtain $f_{P^\pi} \succeq f_{V^\pi}$.

**Case 2**. The reward function $R$ depends on both state $s \in S$ and action $a \in A$, we derive the value function:

$$
V^\pi(s) = \sum_a \pi(a \mid s) \left( R(s, a) + \sum_{s'} P(s' \mid s, a) V^\pi(s') \right).
\tag{9}
$$

If $f_{P^\pi}(\pi_i) = f_{P^\pi}(\pi_j)$, with the Definition 3, we have $\forall s, s' \in S, P^{\pi_i}(s' \mid s) = P^{\pi_j}(s' \mid s)$. Unlike the Eq.7, $\pi_i$ may not be equivalent to $\pi_j$ regarding the abstraction criterion of value irrelevance. Therefore, the partial ordering ($f_{P^\pi} \succeq f_{V^\pi}$) is not obtained under Case 2. Nevertheless, Case 1 is fairly standard across a broad set of real world RL problems. □

*Remark* A.1 (More discussions on $R(s, a)$ and $R(s)$). For Case 2 discussed above, i.e., the reward function $R$ depends on both state $s \in S$ and action $a \in A$, we can further separate it into two categories according to our knowledge on real-world decision-making problems:

- In our first category, the dependence of $R$ on state and action is due to the consequence of $s, a$ in leading the decision system into the specific new state $s'$, i.e., $R(s, a) = \sum_{s'} P(s'|s, a)U(s')$ where $U(s')$ is the utility of $s'$ (we adopt the expectation form for the convenience of discussion). In the cases that fall into this category, it can be easy to re-define the reward function by the utility function, i.e., $R(s, a) = U(s)$ for any $a \in A$. With such a conversion, we can also obtain $f_{P^\pi} \succeq f_{V^\pi}$ in these cases.

- Our second category covers the exclusive cases of the first category. For example, consider an environment, where two actions $a_1, a_2$ lead to the same new state $s'$ from state $s$ but gain different rewards. In such cases, the reward is independent on the dynamics caused by $s, a$. We consider that such cases are minority in the ones of interest.

## A.2  OTHER POLICY ABSTRACTIONS

In this paper, we also propose three other policy abstractions in Definition 6. Before introducing the definitions of additional policy abstraction, we make some necessary notations. We use $d^{\pi,k}(\cdot)$ to denote the distribution of state when policy $\pi$ performs $k$ steps from initial states regarding the initial state distribution $\rho_0$. The discounted state visitation distribution from initial states regarding $\rho_0$ is defined as $d^\pi(s') = (1 - \gamma) \sum_{t=0}^\infty \gamma^t d^{\pi,t}(s')$ for any $s' \in S$. Additionally, we use $d_s^{\pi,k}(\cdot)$ to

denote the distribution of state when policy $\pi$ performs $k$ steps from any states $s$. The discounted state visitation distribution from any state $s$ is defined as $d_s^\pi(s') = (1-\gamma)\sum_{t=0}^\infty \gamma^t d_s^{\pi,t}(s')$ for any $s' \in S$.

**Definition 6.** *Given an MDP $\langle S, A, P, R, \gamma \rangle$ and a ground policy space $\Pi$, for any two policies $\pi_i, \pi_j \in \Pi$, we define four more policy abstractions as follows:*

1. *An influence-irrelevance abstraction ($f_{d_s^\pi}$) is such that for all $s, s' \in S$, $f_{d_s^\pi}(\pi_i) = f_{d_s^\pi}(\pi_j)$ implies that $d_s^{\pi_i}(s') = d_s^{\pi_j}(s')$.*

2. *An influence-irrelevance abstraction ($f_{d^\pi}$) is such that for all $s \in S$, $f_{d^\pi}(\pi_i) = f_{d^\pi}(\pi_j)$ implies that $d^{\pi_i}(s) = d^{\pi_j}(s)$.*

3. *A value-irrelevance abstraction ($f_{J^\pi}$) is such that $f_{J^\pi}(\pi_i) = f_{J^\pi}(\pi_j)$ implies that $\mathbb{E}_{s_0 \sim \rho_0}[V^{\pi_i}(s_0)] = \mathbb{E}_{s_0 \sim \rho_0}[V^{\pi_j}(s_0)]$.*

4. *A value-irrelevance abstraction ($f_{Z^\pi}$) is such that for all $s \in S$, $f_{Z^\pi}(\pi_i) = f_{Z^\pi}(\pi_j)$ implies that $Z^{\pi_i}(s) = Z^{\pi_j}(s)$.*

Similarly, we prove how the newly proposed policy abstractions are related to other abstractions we introduce in the main body of the paper.

**Theorem A.2** (Partial Ordering ($\succeq$)). *Under the Definition 3, 5 and 6, we have ❶ $f_\pi \succeq f_{P^\pi} \succeq f_{d_s^\pi} \succeq f_{d^\pi} \succeq f_{J^\pi}$; ❷ $f_\pi \succeq f_{P^\pi} \succeq f_{d_s^\pi} \succeq f_{V^\pi} \succeq f_{J^\pi}$. An illustration is provided below:*

$$f_\pi \quad \succeq \quad f_{P^\pi} \quad \succeq \quad f_{d_s^\pi} \quad \succeq \quad f_{d^\pi}$$

$$\text{I}\curlyvee \qquad\qquad \text{I}\curlyvee$$

$$f_{V^\pi} \quad \succeq \quad f_{J^\pi}$$

*Proof.* The partial ordering ($\succeq$) satisfies transitivity, thus, let us prove the TheoremA.2 one by one in the following.

❶ $f_{P^\pi} \succeq f_{d_s^\pi}$. Given an MDP $M$, two policies $\pi_i, \pi_j \in \Pi$. If $f_{P^\pi}(\pi_i) = f_{P^\pi}(\pi_j)$, with the Definition 3, we have $\forall s, s' \in S, P^{\pi_i}(s' \mid s) = P^{\pi_j}(s' \mid s)$. Since the initial state distribution are the same, and $\pi_i, \pi_j$ follow two identical Markov chains, we derive $f_{P^\pi} \succeq f_{d_s^\pi}$.

❷ $f_{d_s^\pi} \succeq f_{d^\pi}$. Given an MDP, two policies $\pi_i, \pi_j \in \Pi$. If $f_{d_s^\pi}(\pi_i) = f_{d_s^\pi}(\pi_j)$, with the Definition 6, we have $\forall s, s' \in S, d_s^{\pi_i}(s') = d_s^{\pi_j}(s')$. Furthermore, we derive,

$$\forall s_0 \in \rho_0, s' \in S, \ d_{s_0}^{\pi_i}(s') = d_{s_0}^{\pi_j}(s'). \tag{10}$$

Thus, $f_{d_s^\pi} \succeq f_{d^\pi}$.

❸ $f_{d^\pi} \succeq f_{J^\pi}$. Given an MDP, two policies $\pi_i, \pi_j \in \Pi$. If $f_{d^\pi}(\pi_i) = f_{d^\pi}(\pi_j)$, with the Definition 6, we have $\forall s \in S, d^{\pi_i}(s) = d^{\pi_j}(s)$. When the reward function $R$ depends only on state $s \in S$, we have,

$$\begin{aligned} J(\pi_i) &= (1-\gamma)^{-1}\mathbb{E}_{s \in d^{\pi_i}}[R(s)], \\ J(\pi_j) &= (1-\gamma)^{-1}\mathbb{E}_{s \in d^{\pi_j}}[R(s)]. \end{aligned} \tag{11}$$

Thus, $f_{d^\pi} \succeq f_{J^\pi}$.

❹ $f_{d_s^\pi} \succeq f_{V^\pi}$. Given an MDP, two policies $\pi_i, \pi_j \in \Pi$. If $f_{d_s^\pi}(\pi_i) = f_{d_s^\pi}(\pi_j)$, with the Definition 6, we have $\forall s, s' \in S, d_s^{\pi_i}(s') = d_s^{\pi_j}(s')$. When the reward function $R$ depends only on state $s' \in S$, we have,

$$\begin{aligned} \forall s \in S, \ V^{\pi_i}(s) &= (1-\gamma)^{-1}\mathbb{E}_{s' \in d_s^{\pi_i}}[R(s')], \\ V^{\pi_j}(s) &= (1-\gamma)^{-1}\mathbb{E}_{s' \in d_s^{\pi_j}}[R(s')]. \end{aligned} \tag{12}$$

Thus, $f_{d_s^\pi} \succeq f_{V^\pi}$.

❺ $f_{V^\pi} \succeq f_{J^\pi}$. Given an MDP, two policies $\pi_i, \pi_j \in \Pi$. If $f_{V^\pi}(\pi_i) = f_{V^\pi}(\pi_j)$, with the Definition 3, we have $\forall s \in S, V^{\pi_i}(s) = V^{\pi_j}(s)$. Furthermore, we derive,

$$\begin{aligned} \forall s_0 \in \rho_0, \ V^{\pi_i}(s_0) &= V^{\pi_j}(s_0), \\ \mathbb{E}_{s_0 \sim \rho_0}[V^{\pi_i}(s_0)] &= \mathbb{E}_{s_0 \sim \rho_0}[V^{\pi_j}(s_0)]. \end{aligned} \tag{13}$$

Thus, $f_{V^\pi} \succeq f_{J^\pi}$.

In particular, when the reward function $R$ depends on both state $s \in S$ and action $a \in A$, $\pi_i$ may not be equivalent to $\pi_j$. The $f_{d^\pi} \succeq f_{J^\pi}$ and $f_{d_s^\pi} \succeq f_{V^\pi}$ are not hold. Connecting Definition 3,6 and Theorem A.2, we summarize the properties of different policy abstraction in Table 3. $\qquad\square$

Table 3: Properties of different policy abstraction, including the additional ones introduced in A.2. We use $(+)$ and $(-)$ to denote the higher or lower degree in the same level.

| Abstraction | Abstraction Criterion (for $\pi_1, \pi_2, \forall s, s', a \in S^2 \times A$) | Fineness | Task Relevance |
|---|---|---|---|
| $f_\Theta$ | Policy Parameter Equivalence ($\theta_1 = \theta_2$) | Highest | None |
| $f_\pi$ | Action Distribution Equivalence ($\pi_i(a \mid s) = \pi_j(a \mid s)$) | High | Low |
| $f_{P^\pi}$ | Dynamics Influence Equivalence ($P^{\pi_i}(s'\|s) = P^{\pi_j}(s'\|s)$) | Middle $(+)$ | Middle $(-)$ |
| $f_{d_s^\pi}$ | Dynamics Influence Equivalence ($d_s^{\pi_i}(s') = d_s^{\pi_j}(s')$) | Middle | Middle |
| $f_{d^\pi}$ | Dynamics Influence Equivalence ($d^{\pi_i}(s) = d^{\pi_j}(s)$) | Middle $(-)$ | Middle $(+)$ |
| $f_{V^\pi}$ | Value Function Equivalence ($V^{\pi_i}(s) = V^{\pi_j}(s)$) | Low | High |
| $f_{J^\pi}$ | Value Function Equivalence ($\mathbb{E}_{s_0 \sim \rho_0}[V^{\pi_i}(s_0)] = \mathbb{E}_{s_0 \sim \rho_0}[V^{\pi_j}(s_0)]$) | Low $(-)$ | High $(+)$ |
| $f_0$ | Triviality (taking all policies as the same) | Lowest | None |

Table 4: A taxonomy of prior policy abstractions under our policy abstraction theory.

| Prior Policy Representation | Abstraction Criterion | Related Policy Abstraction |
|---|---|---|
| Vectorized Network Parameters(Faccio et al., 2020) | Policy Parameter Equivalence | $f_\Theta$ |
| Contrastive OPR/SPR(Tang et al., 2020) | Policy Instance Contrast | $f_{CL}$ |
| Policy Recovery OPR/SPR(Tang et al., 2020) | Action Distribution Prediction | $f_\pi$ (Def. 3) |
| End-to-End OPR/SPR(Tang et al., 2020) | Value Function Prediction | $f_{V^\pi}$ (Def. 3) |
| Network Fingerprint(Harb et al., 2020) | Action Distribution Similarity | $f_\pi$ (Def. 3) |
| Behavior Embedding (State)(Pacchiano et al., 2020) | Final State Similarity | $f_{d^\pi}$ (Def. 6) |
| Behavior Embedding (Action)(Pacchiano et al., 2020) | Action Distribution Similarity | $f_\pi$ (Def. 3) |
| Behavior Embedding (Reward)(Pacchiano et al., 2020) | Return Similarity | $f_{J^\pi}$ (Def. 6) |
| Generative Representation(Grover et al., 2018) | Action Distribution Prediction | $f_\pi$ (Def. 3) |
| Discriminative Representation(Grover et al., 2018) | Policy Instance Contrast | $f_{CL}$ |
| $\alpha$-compression (Mutti et al., 2021) | Action Distribution & Dynamics Influence Similarity | $f_\pi$ (Def. 3) & $f_{d^\pi}$ (Def. 6) |
| Policy Supervector(Kanervisto et al., 2020) | Dynamics Influence Similarity | $f_{d^\pi}$ (Def. 6) |

## B  TAXONOMY OF PRIOR POLICY ABSTRACTIONS UNDER OUR THEORY

Closely related to our work, (Harb et al., 2020) adopts policy fingerprints as differentiable policy representation obtained by concatenating the distribution of actions of policy in a set of key states. Obviously, it's more concerned with policy distribution information. (Kanervisto et al., 2020) make use of *Gaussian mixture models* to learn policy supervectors (policy representation), which characterizes agents' behaviour by the distribution of states they visit. Pacchiano et al. (2020) define a Behavioral Embedding Map (BEM) with different implementation options and propose using Wasserstain distance in the latent space induced by BEM to measure the similarity among policies. Like us, (Tang et al., 2020) also employs neural network-based policy parameters as policy data, but it utilize policy distribution reconstruction and contrast learning principles to learn policy representation. The principle of contrastive learning itself is a general learning principle irrelevant to the learning task. Therefore, the contrast of instantiation makes its abstraction level low. Compared with these two works, (Faccio et al., 2020) is a simpler and more direct way of constructing policy representation, because it directly compress the policy parameters based on neural network into vectors and regards it as a form of policy representation. In fact, the policy space may be redundant due to the existence of some policies that induce similar behaviors. It provides feasibility for policy abstraction and policy space compression (Mutti et al., 2021). For instance, Mutti et al. (2021) study policy space compression by formulating a Set Covering problem with Rényi Divergence of discounted state-action distribution of policies as a metric. In addition, in a multi-agent system, (Grover et al., 2018) also converts opponent modeling into an opponent policy representation learning problem and attempts to distinguish opponents by their policy representations. In Table 4, we categorize these works under our abstractions.

## C  CASE ILLUSTRATION OF POLICY ABSTRACTION

To compare these policy abstractions in a quantitative view, we demonstrate how the distances of two policies (blue and greed in Fig. 3) measured by the corresponding policy metrics differ in several Gridworld MDPs. We borrow *Distinct Policies*, *Doorway* from (Kanervisto et al., 2020) and design a new environment, *Key Action* for simple prototypes of environments with different features. All three environments are *5×5* Gridworld, with the starting state and goal in the lower left and upper right grid, respectively. The discrete action space is {*up, down, left, right*}. We obtain the estimated value $v(\cdot)$ of each state by estimating `1 - [expected number of step to goal when starting from a given state]`. About the *Key Action*, the agent obtain a positive reward only if it chooses *right* at the key state (i.e., marked by the red box in Fig. 3). Every agent rollouts 10k episodes for comparing different policy metrics.

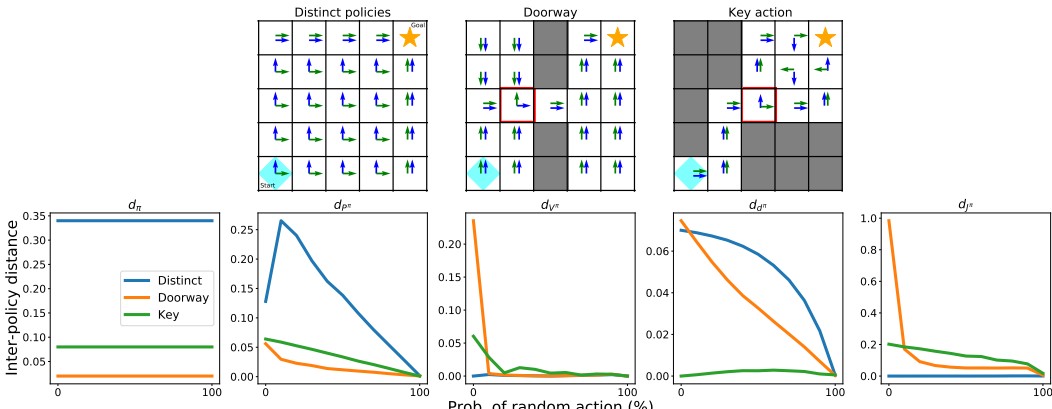

Figure 3: Policy comparison with different policy metrics in Gridworld. *Top Panel:* The illustration of three Gridworld MDPs and two deterministic policies (blue and green). *Bottom Panel:* The distance curves of the two policies measured by $d_\pi, d_{P^\pi}, d_{V^\pi}, d_{d^\pi}, d_{J^\pi}$ ($y$-axi), against the stochasticity of the environment ($x$-axi).

To be specific, a policy is represented by tensor $\pi \in \mathbb{R}^{5\times5\times4}$ and the distribution-irrelevance metric is defined as $d_\pi(\pi, \hat{\pi}) = \mathbb{E}_{\pi_{i,j,k}\in\pi, \hat{\pi}_{i,j,k}\in\hat{\pi}}\left[|\pi_{i,j,k} - \hat{\pi}_{i,j,k}|\right]$. Similarly, the state transition dynamics induced by policy is represented by tensor $P^\pi \in \mathbb{R}^{5\times5\times5}$ and the influence-irrelevance metric is defined as $d_{P^\pi}(\pi, \hat{\pi}) = \mathbb{E}_{P^\pi_{i,j,k}\in P^\pi, P^{\hat{\pi}}_{i,j,k}\in P^{\hat{\pi}}}\left[\left|P^\pi_{i,j,k} - P^{\hat{\pi}}_{i,j,k}\right|\right]$. Different from the $d_{P^\pi}$ sampling-based estimation, we leverage the dynamic programming to learn the value function $V^\pi \in \mathbb{R}^{5\times5}$. Thus, the value-irrelevance metric is defined as $d_{V^\pi}(\pi, \hat{\pi}) = \mathbb{E}_{V^\pi_{i,j}\in V^\pi, V^{\hat{\pi}}_{i,j}\in V^{\hat{\pi}}}\left[\left|V^\pi_{i,j} - V^{\hat{\pi}}_{i,j}\right|\right]$. In addition, we also compare the two other policy abstractions proposed in Appendix A.2. Among them, we estimate the discounted state visitation distribution $d^\pi(s)$ via dividing visits to a state $s$ by a total number of interactions (i.e., $d_{d^\pi}(\pi, \hat{\pi}) = \mathbb{E}_{d^\pi_{i,j}\in d^\pi, d^{\hat{\pi}}_{i,j}\in d^{\hat{\pi}}}\left[|d^\pi_{i,j} - d^{\hat{\pi}}_{i,j}|\right]$). Simply and naturally, based on the definition of $f_{J^\pi}$ in Definition 6, we have $d_{J^\pi}(\pi, \hat{\pi}) = |\mathbb{E}_{s_0\sim\rho_0}\left[V^\pi(s_0)\right] - \mathbb{E}_{s_0\sim\rho_0}\left[V^{\hat{\pi}}(s_0)\right]|$. Fig. 3 shows the illustrations and the results of the five policy metrics. The $d_{d^\pi}$ and $d_{J^\pi}$ are newly added results compared to the original paper. We observe that the $d_{d^\pi}$ cannot indicate the difference in dynamics between the two policies in *Key Action*. Like the $d_{V^\pi}$, the $d_{J^\pi}$ measures the difference in the outcomes of the two policies but shows the poor robustness as the increase of stochasticity.

# D  ADDITIONAL DISCUSSIONS

## D.1  ESTIMATING POLICY METRICS VIA JEFFREYS DIVERGENCE

In simple MDPs where the state-action space is finite, we calculate the frequency distribution (i.e., $\widetilde{\pi}, \widetilde{P}^\pi, \widetilde{Z}^\pi$) using sufficient samples as an estimate of the exact probability distribution (i.e., $\pi, P^\pi, Z^\pi$). Then we use the Jeffreys Divergence (Jeffreys, 1946) between empirical distributions as policy metrics (i.e., $d_\pi(\cdot, \cdot), d_{P^\pi}(\cdot, \cdot)$, and $d_{V^\pi}(\cdot, \cdot)$),

$$
\begin{aligned}
d_\pi(\pi_i, \pi_j) &\approx D_{KL}\left(\widetilde{\pi}_i\|\widetilde{\pi}_j\right) + D_{KL}\left(\widetilde{\pi}_j\|\widetilde{\pi}_i\right), \\
d_{P^\pi}(\pi_i, \pi_j) &\approx D_{KL}\left(\widetilde{P}^{\pi_i}\|\widetilde{P}^{\pi_j}\right) + D_{KL}\left(\widetilde{P}^{\pi_j}\|\widetilde{P}^{\pi_i}\right), \\
d_{V^\pi}(\pi_i, \pi_j) &\approx D_{KL}\left(\widetilde{Z}^{\pi_i}\|\widetilde{Z}^{\pi_j}\right) + D_{KL}\left(\widetilde{Z}^{\pi_j}\|\widetilde{Z}^{\pi_i}\right).
\end{aligned}
\tag{14}
$$

# E  DETAILS ON APPLYING POLICY ABSTRACTIONS TO POLICY OPTIMIZATION

The policy optimization experiments are run on a single NVIDIA GeForce GTX 2080Ti GPU. Our codes are implemented with Python 3.7.13 and Torch 1.11.0.

Table 5: Trust-region constraint value choices of different Methods. We use '–' to denote the 'not applicable' situation.

| Methods | Trust-region threshold ($\sigma$) |
|---------|-----------------------------------|
| Vanilla-PO | – |
| Max TV | $\{0.001, 0.005, 0.01, 0.05, 0.1, 0.2, 0.3, 0.4, 0.5\}$ |
| Gaussian | $\{0.5, 1.0, 2.0, 3.0, 5.0, 10.0, 15.0, 20.0\}$ |
| Supervector | $\{0.01, 0.05, 0.1, 0.15, 0.2, 0.3, 0.4, 0.5\}$ |
| TRPO-$f_\pi$ | $\{0.05, 0.1, 0.2, 0.3, 0.4, 0.5\}$ |
| TRPO-$f_{P\pi}$ | $\{0.05, 0.1, 0.2, 0.3, 0.4, 0.5\}$ |
| TRPO-$f_{V\pi}$ | $\{0.05, 0.1, 0.5, 1.0, 2.0, 5.0\}$ |

### E.1 TRUST-REGION POLICY OPTIMIZATION

In this experiment, we adopt a $N$-dimensional Gridworld MDP provided by (Kanervisto et al., 2020), where the position coordinates of the agent form the state space. At each state, the agent choose one of $N$ actions corresponding $N$ directions. In the experiments, we set $N$=5 and only one action of $N$ action move the agent forward. The agent is rewarded with +1 reward for taking the correct action and the maximum length of each episode is 25. The agent uses a two-layer network with 16 units and tanh-activations each.

For the policy optimization, we rewrite the implementation of the policy metrics in the code of (Kanervisto et al., 2020). The learning agent checks if the difference between old and new policies measured by the policy metrics are larger than trust-region threshold $\sigma$. If the threshold is exceeded, we stop updating the policy with current samples and move on to collect new samples for the next policy update. The sample size is 4096 and the policy is updated for 100 mini-batches of 64 items over the collected samples, or until constraint prevents updates.

We compare with existing related methods, i.e.,Vanilla-PO, Total Variation Divergence(Max TV), Gaussian and Supervector, and follow hyperparameters in (Kanervisto et al., 2020). Among them, Vanilla-PO means updating policy 100 mini-batches with no trust-region constraint; the Total Variation Divergence measures the maximum amount of how much probability of taking any single action (in any state) can change and is defined as $d(\pi_i, \pi_j) = \max_s \frac{1}{2} \sum_a |\pi_i(a \mid s) - \pi_j(a \mid s)|$; the Gaussian refers to fitting a multivariate, diagonal Gaussian on collected states from 5 trajectories for old and new policy, respectively and measures Jeffreys Divergence between old and new policy; the Supervector calculates the upper bound of KL-divergence between old and new policy supervectors, which are obtained via fitting a four-component UBM on collected states from 5 trajectories for corresponding policy. To ensure a fair comparison, we estimate the proposed policy metrics $d_\pi$, $d_{P\pi}$, $d_{V\pi}$ on collected 5 trajectories from old and new policy. All approaches repeat ten times for same environment and the search space of trust-region threshold for them are shown in Table 5.

Fig. 4(a) reports the empirical results for different methods using the corresponding optimal thresholds which are selected based on the largest AUC (i.e., Area Under The Curve). Overall, the TRPO-$f_\pi$ and Max TV are better than others, mainly because they concern the action distribution at interested states. The Gaussian and Supervector are concerned with the state visitation distribution induced by the policy, which are specific instances of our proposed dynamics influence abstraction. Thus, as with $f_{P\pi}$ and $f_{V\pi}$, they ignore the differences of policies in action distribution.

### E.2 DIVERSITY-GUIDED EVOLUTION STRATEGY

In the experiments, we adopt the Point environment with deceptive rewards using MuJoCo simulator. The deception comes from a barrier, which misleads the agent to move directly forward leading to a suboptimal policy. For the Point environment, the state and action are represented by a 6 dimensional vector and 2 dimensional vector, respectively. At each timestep, the agent is penalized for its distance from a given goal, and we limit episode length to 50 steps. To reduce the number of policy parameters and thus the time cost, we use a Toeplitz policy (Choromanski et al., 2018) often used for ES algorithms. For DGES, we set the population size to 50 (i.e., $N = 50$), and estimate the proposed policy metrics $d_\pi$, $d_{P\pi}$, $d_{V\pi}$ on one episode from the currently policy and the ancestor policy.

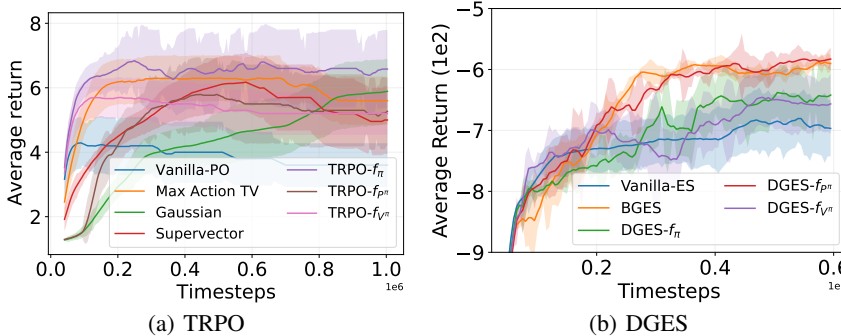

Figure 4: Performance of different methods in: *(a)* Trust-Region Policy Optimization (TRPO); and *(b)* Diversity-Guided Evolution Strategy (DGES). Results are the mean and half a standard deviation (shaded) over 10 and 5 trials for TRPO and DGES respectively.

We compare with state-of-the art method on the Point environment for exploration: BGES (Pacchiano et al., 2020), which uses the terminal state as a policy representation and compare two policies by Wasserstein distances. The hyperparameters associated with the BGES remain consistent with the original paper. From the Fig. 4(b), we observe our approach DGES-$f_{P^\pi}$ is comparable to the BGES (Pacchiano et al., 2020) and performs better than the DGES-$f_\pi$ and DGES-$f_{V^\pi}$. It illustrates the influence-irrelevance metric $d_{P^\pi}$ is a *sweet* intermediate point, keeping the ability of distinguishing the two policies.

# F    DETAILS ON APPLYING POLICY ABSTRACTIONS TO OFF-POLICY EVALUATION

The off-policy evaluation experiments are run on a single NVIDIA GeForce GTX 2080Ti GPU. Our codes are implemented with Python 3.6.13 and tensorflow 2.2.0.

## F.1    COLLECT POLICY DATA FOR OFF-POLICY EVALUATION

**Benchmark Environments.** We conduct our experiments on OpenAI Gym continuous control tasks and detailed description is below.

- *InvertedDoublePendulum-v2 (IDP-v2)*: Balance a pole on a pole on a cart. The agent gets a reward for every timestep that the pendulum has not fallen off the cart.
- *LunarLanderContinuous-v2 (LLC-v2)*: Navigate a lander to its landing pad. Landing pad is always at coordinates (0,0). Reward for moving from the top of the screen to landing pad and zero speed is about 100..140 points. If lander moves away from landing pad it loses reward back. Episode finishes if the lander crashes or comes to rest, receiving additional -100 or +100 points. Each leg ground contact is +10. Firing main engine is -0.3 points each frame. Solved is 200 points.

**Policy Data Collection.** We collect policies in two stages. The first stage, we train the PPO (Schulman et al., 2017a) agent for 1M steps and store a checkpoint each 10 updates of the policy. Then, we evaluate each checkpoint by 10 rollouts. To be specific, for the ppo agent, the update frequency of critic is 5 per epoch on two environments, and the update frequency of actor is 2,5,10 per epoch on IDP-v2 and 5,10,20 per epoch on LLC-v2, respectively. The ppo agent with each update frequency is trained using 20 random seeds. The second stage, we divide 50 intervals $I_\tau(\tau = 1, \cdots, 50)$ depending on the performance range of collected policy set $\mathcal{I}$. Naturally, each interval contains a certain number of policies. Then, we randomly select $K(K = 40)$ policies from each interval. Finally, we collect $B(B = 200)$ trajectories of data per selected policy. Further, for each policy, we calculate the average return $\bar{G}_0$ over the 200 trajectories, and randomly sample state-action pairs $\{(s_j, a_j)\}_{j=1}^{j=m}$, state-next state pairs $\{(s_j, s'_j)\}_{j=1}^{j=m}$, state-value pairs $\{(s_j, G_j)\}_{j=1}^{j=m}$. The pseudo-code for policy data collection is presented in Algorithm1.

In the experiment, we construct a policy dataset for IDP-v2 and LLC-v2, respectively, and each policy dataset contains 2000 policies. Fig. 6 shows the histograms of the two policy datasets, where the

---

**Algorithm 1** Collect Policy Data for Off-policy Evaluation

---

**Input**: Policy set $\mathcal{I} = \{I_1, I_2, \cdots, I_\tau\}$ collected based on PPO algorithm; the number $\tau$ of policy performance intervals; the number $K$ of policies selected per interval; the rollouts $B$ and policy dataset $\mathbb{D} \leftarrow \emptyset$

1:  for $\tau = 1, 2, \cdots$ do
2:      Randomly select $K$ policies in $I_\tau$
3:      for $k = 1$ to $K$ do
4:          $R \leftarrow \emptyset, T \leftarrow \emptyset$
5:          for rollout $b = 1$ to $B$ do
6:              Sample Monte-Carlo return $G_0$ using policy $\pi_k$
7:              Collect data with $\pi_k$ in real environments: $\{(s, a, r, s')\}_b$
8:              $R \leftarrow R \cup \{G_0\}, T \leftarrow T \cup \{(s, a, r, s')\}_b$
9:          end for
10:         $\mathbb{D} \leftarrow \mathbb{D} \cup \{\theta_{\pi_k}, R, T\}, \theta_{\pi_k}$ represents the parameters of policy $\pi_k$
11:     end for
12:  end for

**Output**: $\mathbb{D}$

---

---

**Algorithm 2** Off-policy Evaluation

---

**Input**: Training policy dataset $\mathbb{D}^{train}$; the data of the policy $\pi$ consisting of policy parameters $\theta_\pi$, state-action pairs $\{(s_j, a_j)\}_{j=1}^{j=m}$, state-next state pairs $\{(s_j, s_j')\}_{j=1}^{j=m}$ and state-value pairs $\{(s_j, G_j)\}_{j=1}^{j=m}$; the expected return $\bar{G}_0$

**Initialize**: The policy evaluation function $\mathbb{V}_\beta$ with parameters $\beta$
**Initialize**: The policy representation function $f_\psi$ with parameters $\psi$

1:  Calculate $d_\pi, d_{P^\pi}$, and $d_{V^\pi}$ for all $(\pi_i, \pi_j) \in \mathbb{D}^{train}$          ▷ see Section 4.2
2:  for iteration t = 0, 1, 2, $\cdots$ do
3:      Sample a mini-batch $N$ of policy data $\mathcal{B}$ from $\mathbb{D}^{train}$
4:      Calculate the alignment loss $\mathcal{L}_{AL}$
5:      $\mathcal{L}_{AL}(\psi) = \mathbb{E}_{\pi_i, \pi_j \in \mathcal{B}} \left[ (\|f_\psi(\theta_{\pi_i}) - f_\psi(\theta_{\pi_j})\|_2 - \eta \cdot d_*(\pi_i, \pi_j))^2 \right], d_* \in \{d_\pi, d_{P^\pi}, d_{V^\pi}\}$
6:      Calculate the evaluation loss $\mathcal{L}_{EL}$
7:      $L_{EL}(\beta, \psi) = \mathbb{E}_{\pi_i \in \mathcal{B}} \left[ (\mathbb{V}_\beta(f_\psi(\theta_{\pi_i})) - \bar{G}_0)^2 \right]$
8:      Update parameters $\psi, \beta$ to minimize $\mathcal{L}_{AL}$ and $\mathcal{L}_{EL}$
9:  end for

**Output**: Parameters $\psi$ of the policy representation function $f_\psi$; parameters $\beta$ the policy evaluation function $\mathbb{V}_\beta$

---

x-axis is the average return of policies and the y-axis is the number of policies. As shown in the Fig. 6, the two policy datasets basically satisfy the diversity and balance of policies, which allows us to design a series of Off-Policy Evaluation (OPE) scenarios.

### F.2 OFF-POLICY EVALUATION

With regard to off-policy evaluation tasks, in this work, we adopt a policy evaluation function (Harb et al., 2020) representing the expected return of a policy. Naturally, the policy evaluation function can be trained based on sampled training policies and then generalize to unseen policies. The policy evaluation loss $\mathcal{L}_{EL}$ can be formalized as,

$$L_{EL}(\beta, \psi) = \mathbb{E}_{\pi \in \mathbb{D}^{train}} \left[ (\mathbb{V}_\beta(f_\psi(\theta_\pi)) - \bar{G}_0)^2 \right], \tag{15}$$

where $\mathbb{V}_\beta(\cdot)$ parameterized by $\beta$ denotes the policy evaluation function. $f_\psi(\cdot)$ parameterized by $\psi$ is the policy representation function which takes the policy parameters $\theta_\pi$ as input. $\mathbb{D}^{train}$ denotes the training policy dataset sampled from the collected policy dataset $\mathbb{D}$. Below, we provide the implementation details and pseudo-code (Algorithm 2) for off-policy evaluation.

**Network Structure.** The Fig. 5 illustrates the policy encoder (LPE) we used. The Table 6 shows the structure of the policy network and the policy evaluation network. As shown in Table 6, we use a two-layer feed-forward neural network of 32 and 32 hidden units with ReLU activation (except for the output layer) for the policy network. In this paper, we learn a policy representation (pr) for a

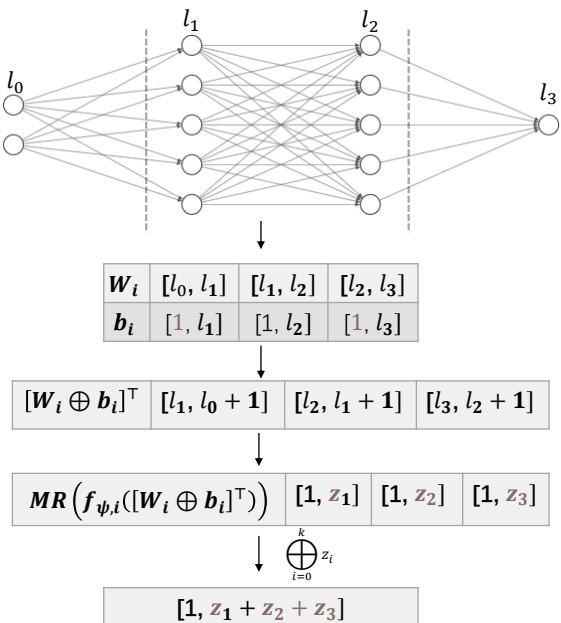

Figure 5: An illustration for Layer-wise Permutation-invariant Encoder (LPE). $l_i$ denotes the numbers of hidden units for the hidden layers. The main idea is to perform permutation-invariant transformation for inner-layer weights and biases for each layer first and then concatenate encoding of layers.

Table 6: Structure of policy network and policy evaluation network.

| Layer | Policy Network ($\pi(a|s)$) | Policy Evaluation Network ($\mathbb{V}(\cdot)$) |
|---|---|---|
| Fully Connected | (state dim, 32) | (pr dim, 128) |
| Activation | ReLU | ReLU |
| Fully Connected | (32, 32) | (128, 128) |
| Activation | ReLU | ReLU |
| Fully Connected | (32, action dim) | (128, 1) |
| Activation | tanh | None |

RL policy from its network parameters. Like the policy network, the policy evaluation network also uses a two-layer feed-forward neural network, where the input to the policy evaluation network is the policy representation and the output is the predicted value of the expected return for the policy.

**Hyperparameter.** In the Table 7, we discuss the policy representation dimension, the number of hidden units of the policy evaluation network, and the batch size for training policy evaluation network. For a fair comparison, we perform a 10-fold cross-validation weak generalization experiment with 20% sampling ratio based on the $f_{RE}$ and IDP-v2 to search for best hyperparameters. For all methods and environments, the batch size, the number of hidden units and the policy representation dimension are 256, 128 and 256, respectively. Additionally, we report the common hyperparamters of off-policy evaluation experiments in Table 8.

## G    OTHER EXPERIMENTAL RESULTS

### G.1    COMPLETE EXPERIMENTAL RESULTS OF DIFFERENT GENERALIZATION SCENARIOS

In the paper, we design two OPE generalization settings, namely weak generalization and strong generalization. For each of the generalization settings, we further construct generalization scenarios with different sampling ratios. As with the Table 2, the results of the generalization experiments on the 40% and 80% sampling ratio are presented in the Table 9 and the Table 10, respectively. Not surprisingly, the performance of all methods improves as the amount of training data increases. In

Table 7: Experimental results of $f_{RE}$ for IDP-v2 with different hyperparameters. The minimum value is highlighted. Results(×1e-2) are the mean ± a standard deviation over 10 trials (for weak generalization with 20% sampling ratio).

| batch size | hidden units | representation dimension | | | | | |
| --- | --- | --- | --- | --- | --- | --- | --- |
| | | 64 | | 128 | | 256 | |
| | | T-error | G-gap | T-error | G-gap | T-error | G-gap |
| 64 | 64 | 0.64±0.11 | 0.42±0.10 | 0.59±0.05 | 0.39±0.06 | 0.55±0.05 | 0.34±0.05 |
| 64 | 128 | 0.59±0.07 | 0.40±0.08 | 0.56±0.06 | 0.36±0.09 | 0.59±0.07 | 0.39±0.09 |
| 64 | 256 | 0.61±0.09 | 0.42±0.10 | 0.57±0.06 | 0.37±0.07 | 0.59±0.09 | 0.37±0.09 |
| 128 | 64 | 0.62±0.06 | 0.40±0.07 | 0.58±0.09 | 0.37±0.09 | 0.57±0.06 | 0.37±0.07 |
| 128 | 128 | 0.64±0.06 | 0.43±0.07 | 0.55±0.06 | 0.36±0.09 | 0.57±0.06 | 0.39±0.08 |
| 128 | 256 | 0.58±0.07 | 0.38±0.07 | 0.56±0.06 | 0.37±0.08 | 0.55±0.05 | 0.37±0.08 |
| 256 | 64 | 0.69±0.10 | 0.38±0.10 | 0.64±0.09 | 0.38±0.09 | 0.60±0.06 | 0.36±0.08 |
| 256 | 128 | 0.65±0.09 | 0.38±0.07 | 0.64±0.07 | 0.41±0.10 | **0.54±0.05** | **0.34±0.05** |
| 256 | 256 | 0.65±0.09 | 0.41±0.10 | 0.59±0.07 | 0.38±0.09 | 0.55±0.07 | 0.35±0.10 |

Table 8: A comparison of common hyperparameter choices of algorithms. We use '-' to denote the 'not applicable' situation.

| Hyperparameter | $f_\Theta$ | $f_{RE}$ | $f_{EL}$ | $f_{CL}$ | $f_\pi, f_{P^\pi}, f_{V^\pi}$ |
| --- | --- | --- | --- | --- | --- |
| Evaluation Model Learning Rate | $1 \cdot 10^{-3}$ | $1 \cdot 10^{-3}$ | $1 \cdot 10^{-3}$ | $1 \cdot 10^{-3}$ | $1 \cdot 10^{-3}$ |
| Representation Model Learning Rate | $1 \cdot 10^{-3}$ | $1 \cdot 10^{-3}$ | $1 \cdot 10^{-3}$ | $1 \cdot 10^{-3}$ | $1 \cdot 10^{-3}$ |
| Optimizer | Adam | Adam | Adam | Adam | Adam |
| Batch Size | 256 | 256 | 256 | 256 | 256 |
| Policy Representation dim | - | 256 | 256 | 256 | 256 |
| Evaluation Model Update Epoch (20%) | 10k | 10k | 10k | 10k | 10k |
| Evaluation Model Update Epoch (40%) | 50k | 50k | 50k | 50k | 50k |
| Evaluation Model Update Epoch (80%) | 100k | 100k | 100k | 100k | 100k |
| Kernel mu | - | - | - | - | 2.0 |
| Kernel number | - | - | - | - | 5 |
| Sample Size $m$ | - | - | - | - | 1000 |

the weak generalization experimental scenarios, our methods are better than or comparable to other methods. In the strong generalization experimental scenarios, the proposed method, especially $f_{V^\pi}$, is significantly more robust than the other methods.

## G.2 HOW DIFFERENT POLICY ABSTRACTIONS INTRAPOLATE AND EXTRAPOLATE IN OPE?

To present the results of weak and strong generalization more intuitively, we attach scatter plots of policy evaluation results for two generalization settings with 20% sampling ratio in Fig. 7,8,9,10. To be specific, the horizontal axis of scatter plots represents the true value and the vertical axis is the predicted value. The evaluation results of the training policies (red dots) and target policies (blue dots) are unified in a scatter plot for each method. Experimental results are the best of 10 and 5 trials (for weak and strong respectively). Obviously, the results of the weak generalization setting basically remained near the diagonal, while the strong generalization experiments show an overall underestimation. This is due to the fact that in the strong generalization setting we only sample low-performance policies as training data and the rest (i.e., high-performance policies) are used as test policies.

## G.3 VISUALIZATION OF LEARNED POLICY REPRESENTATIONS

In this work, we further analyze different policy abstractions (i.e., $f_\Theta$, $f_{RE}$, $f_{EL}$, $f_\pi$, $f_{P^\pi}$, $f_{V^\pi}$) by visualizing policy representations. Among them, the $f_\Theta$ denotes directly compressing policy parameters as policy representations. The $f_{RE}$ and $f_{EL}$ refer to learning policy representations by random initial abstraction function and optimized abstraction function by the evaluation loss, respectively. The unsupervised contrastive learning approach ($f_{CL}$) learns policy representations based on the InforNCE (Tang et al., 2020). $f_\pi$, $f_{P^\pi}$, $f_{V^\pi}$ correspond to the three policy abstractions we propose, and the policy representations are learned via the alignment loss.

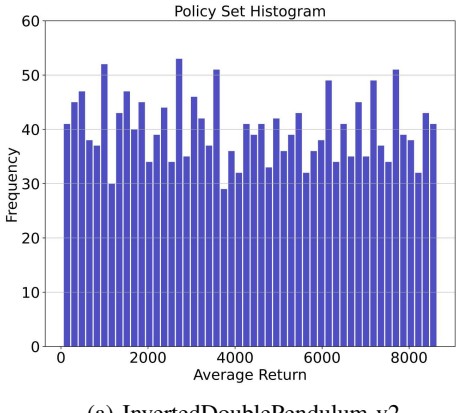
(a) InvertedDoublePendulum-v2

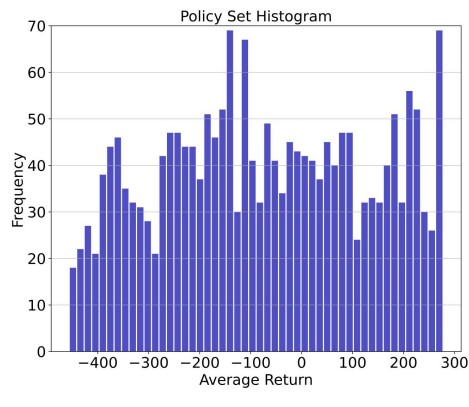
(b) LunarLanderContinuous-v2

Figure 6: Histograms of policy dataset collected from the (a) InvertedDoublePendulum-v2 and (b) LunarLanderContinuous-v2 environments, where the x-axis is the average return of policies and the y-axis is the number of policies.

**Visualization Details.** We perform experiments under two generalization settings with 20% sampling ratio. For a fair comparison, different methods use the same training dataset to optimize the policy representation function. The hyperparameters about training phrase can be found in Table8. We store the last policy representation model for all methods, and compute the representations of training policies and target(unseen) policies. Finally, we plot 2D embedding of each policy representation by the T-SNE (V. d. Maaten & Hinton, 2008). In the visualisation results, each colored point represents a policy and the label of the color bar for all results is the expected return of the policy.

**Visualization Results.** Overall, the policy representations in Fig. 11 are more clearly distinguishable and all of our proposed methods show some degree of discriminability. In contrast, in the strong generalization experiments, the better performing policies are less discriminative as only 20% of the poor performing policies are used as training data. Nevertheless, our proposed value-irrelevance policy abstraction still learns a more compact and discriminative representation. The same results can also be found in the LLC-v2 environment (Fig. 13, 14). Differently, $f_{RE}$ has already obtained a better discriminability on that environment. We consider two possible reasons for this result, one being due to the variability of the different environments, and the other possibly being the superiority of the policy encoder network (LPE) we adopt. To verify the latter, we show the visualization of the unabstracted policy in fig. 15. From the results, it is clear that our policy encoder network is indeed able to perform an effective compression of the policy information and obtain a compact policy representation space.

Table 9: Performance of different policy abstractions in Off-policy Evaluation (OPE). The minimum value for each task is highlighted. Results are the mean $\pm$ a standard deviation over 10 and 5 trials (for *weak* and *strong* with 40% sampling ratio respectively).

| Env | Abstraction | Weak Generalization | | Strong Generalization | |
|---|---|---|---|---|---|
| | | T-error | G-gap | T-error | G-gap |
| IDP-v2 | $f_\Theta$ | $0.0041 \pm 0.0002$ | $0.0018 \pm 0.0005$ | $0.0684 \pm 0.0056$ | $0.0679 \pm 0.0057$ |
| | $f_{RE}$ | $0.0042 \pm 0.0004$ | $0.0021 \pm 0.0005$ | $0.0752 \pm 0.0123$ | $0.0728 \pm 0.0124$ |
| | $f_{EL}$ | $0.0039 \pm 0.0002$ | $0.0015 \pm 0.0005$ | $0.0725 \pm 0.0106$ | $0.0717 \pm 0.0098$ |
| | $f_{CL}$ | $0.0046 \pm 0.0002$ | $0.0025 \pm 0.0004$ | $0.0762 \pm 0.0059$ | $0.0676 \pm 0.0091$ |
| | $f_\pi$ | $\mathbf{0.0037 \pm 0.0002}$ | $0.0014 \pm 0.0005$ | $0.0843 \pm 0.0091$ | $0.0816 \pm 0.0087$ |
| | $f_{P\pi}$ | $\mathbf{0.0037 \pm 0.0002}$ | $0.0014 \pm 0.0005$ | $0.0874 \pm 0.0063$ | $0.0845 \pm 0.0077$ |
| | $f_{V\pi}$ | $0.0037 \pm 0.0003$ | $\mathbf{0.0011 \pm 0.0005}$ | $\mathbf{0.0584 \pm 0.0072}$ | $\mathbf{0.0548 \pm 0.0087}$ |
| LLC-v2 | $f_\Theta$ | $0.0008 \pm 0.0001$ | $0.0007 \pm 0.0001$ | $0.1061 \pm 0.0137$ | $0.1046 \pm 0.0138$ |
| | $f_{RE}$ | $0.0013 \pm 0.0003$ | $0.0011 \pm 0.0003$ | $0.0339 \pm 0.0142$ | $0.0295 \pm 0.0124$ |
| | $f_{EL}$ | $0.0008 \pm 0.0001$ | $0.0007 \pm 0.0001$ | $0.0322 \pm 0.0035$ | $0.0320 \pm 0.0036$ |
| | $f_{CL}$ | $0.0015 \pm 0.0002$ | $0.0013 \pm 0.0002$ | $0.0434 \pm 0.0019$ | $0.0365 \pm 0.0040$ |
| | $f_\pi$ | $\mathbf{0.0008 \pm 0.0001}$ | $0.0006 \pm 0.0001$ | $\mathbf{0.0193 \pm 0.0022}$ | $\mathbf{0.0163 \pm 0.0023}$ |
| | $f_{P\pi}$ | $\mathbf{0.0008 \pm 0.0001}$ | $\mathbf{0.0006 \pm 0.0000}$ | $0.0215 \pm 0.0041$ | $0.0169 \pm 0.0055$ |
| | $f_{V\pi}$ | $\mathbf{0.0008 \pm 0.0001}$ | $0.0006 \pm 0.0001$ | $0.0301 \pm 0.0024$ | $0.0278 \pm 0.0043$ |

Table 10: Performance of different policy abstractions in Off-policy Evaluation (OPE). The minimum value for each task is highlighted. Results are the mean $\pm$ a standard deviation over 10 and 5 trials (for *weak* and *strong* with 80% sampling ratio respectively).

| Env | Abstraction | Weak Generalization | | Strong Generalization | |
|---|---|---|---|---|---|
| | | T-error | G-gap | T-error | G-gap |
| IDP-v2 | $f_\Theta$ | $0.0029 \pm 0.0002$ | $0.0004 \pm 0.0004$ | $0.0181 \pm 0.0000$ | $0.0140 \pm 0.0005$ |
| | $f_{RE}$ | $0.0029 \pm 0.0002$ | $0.0005 \pm 0.0004$ | $0.0117 \pm 0.0002$ | $0.0063 \pm 0.0011$ |
| | $f_{EL}$ | $0.0028 \pm 0.0002$ | $0.0004 \pm 0.0004$ | $0.0135 \pm 0.0017$ | $0.0067 \pm 0.0026$ |
| | $f_{CL}$ | $0.0028 \pm 0.0003$ | $0.0011 \pm 0.0005$ | $0.0100 \pm 0.0009$ | $\mathbf{-0.0072 \pm 0.0004}$ |
| | $f_\pi$ | $0.0028 \pm 0.0001$ | $\mathbf{0.0003 \pm 0.0003}$ | $0.0127 \pm 0.0006$ | $-0.0063 \pm 0.0065$ |
| | $f_{P\pi}$ | $\mathbf{0.0027 \pm 0.0002}$ | $0.0003 \pm 0.0004$ | $0.0158 \pm 0.0010$ | $0.0004 \pm 0.0019$ |
| | $f_{V\pi}$ | $\mathbf{0.0027 \pm 0.0002}$ | $0.0004 \pm 0.0005$ | $\mathbf{0.0082 \pm 0.0027}$ | $-0.0020 \pm 0.0069$ |
| LLC-v2 | $f_\Theta$ | $0.0005 \pm 0.0001$ | $0.0004 \pm 0.0001$ | $0.0102 \pm 0.0012$ | $0.0078 \pm 0.0003$ |
| | $f_{RE}$ | $0.0007 \pm 0.0001$ | $0.0005 \pm 0.0001$ | $0.0127 \pm 0.0036$ | $0.0122 \pm 0.0039$ |
| | $f_{EL}$ | $\mathbf{0.0005 \pm 0.0000}$ | $0.0004 \pm 0.0001$ | $\mathbf{0.0029 \pm 0.0005}$ | $0.0026 \pm 0.0005$ |
| | $f_{CL}$ | $0.0007 \pm 0.0001$ | $0.0006 \pm 0.0001$ | $0.0061 \pm 0.0022$ | $0.0037 \pm 0.0046$ |
| | $f_\pi$ | $\mathbf{0.0005 \pm 0.0000}$ | $\mathbf{0.0003 \pm 0.0000}$ | $0.0035 \pm 0.0001$ | $\mathbf{0.0022 \pm 0.0002}$ |
| | $f_{P\pi}$ | $\mathbf{0.0005 \pm 0.0000}$ | $0.0004 \pm 0.0000$ | $0.0043 \pm 0.0003$ | $0.0037 \pm 0.0003$ |
| | $f_{V\pi}$ | $0.0005 \pm 0.0001$ | $0.0004 \pm 0.0001$ | $0.0032 \pm 0.0012$ | $0.0028 \pm 0.0013$ |

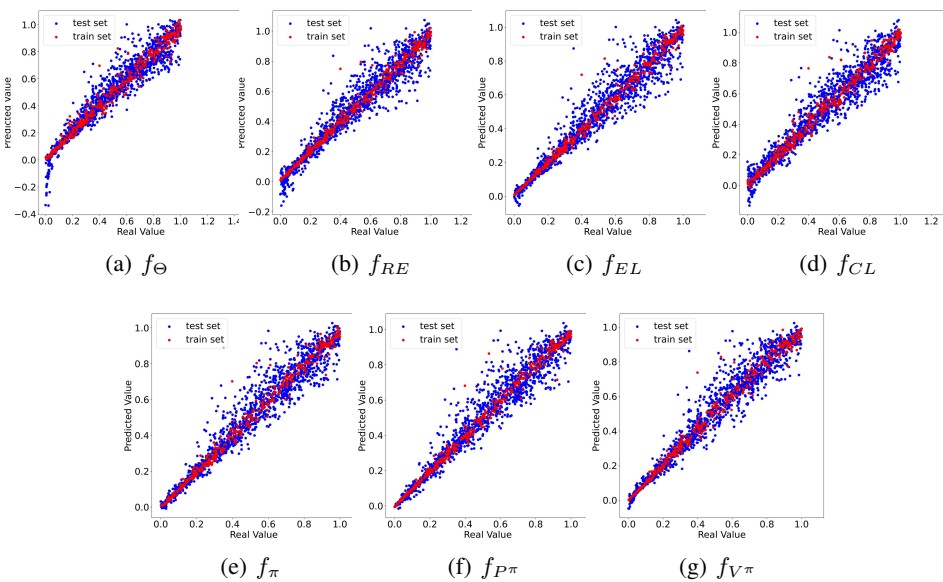

Figure 7: Results of policy evaluation with 20% sampling ratio (Weak Generalization, IDP-v2). The results show the true and predicted values of the train policies (red dots) and target policies (blue dots) at the minimum testing error for policy abstraction methods.

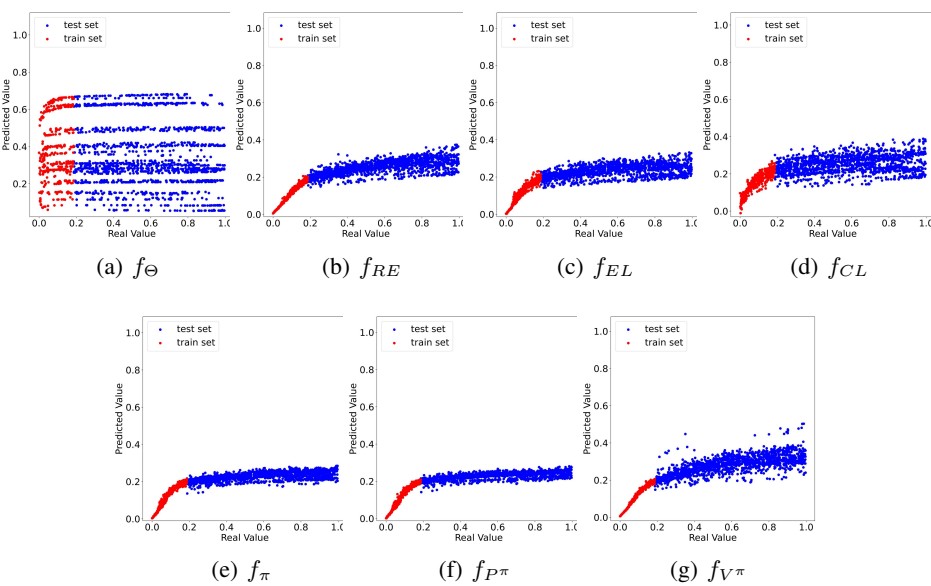

Figure 8: Results of policy evaluation with 20% sampling ratio (Strong Generalization, IDP-v2). The results show the true and predicted values of the train policies (red dots) and target policies (blue dots) at the minimum testing error for policy abstraction methods.

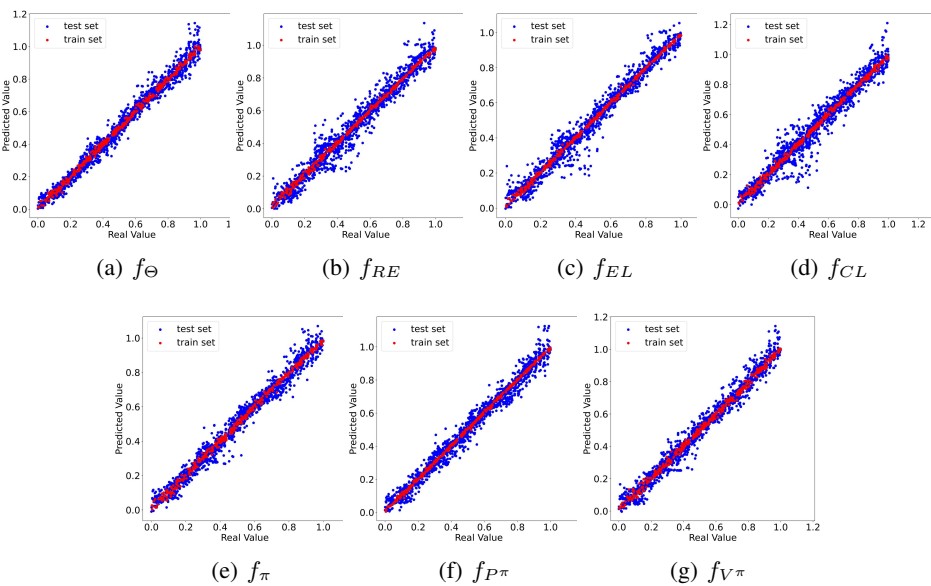

Figure 9: Results of policy evaluation with 20% sampling ratio (Weak Generalization, LLC-v2). The results show the true and predicted values of the train policies (red dots) and target policies (blue dots) at the minimum testing error for policy abstraction methods.

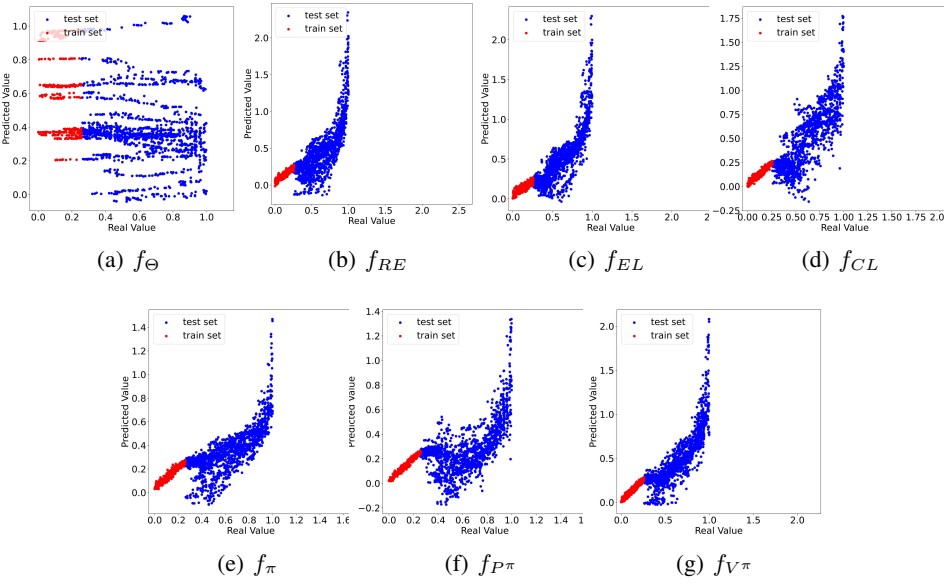

Figure 10: Results of policy evaluation with 20% sampling ratio (Strong Generalization, LLC-v2). The results show the true and predicted values of the train policies (red dots) and target policies (blue dots) at the minimum testing error for policy abstraction methods.

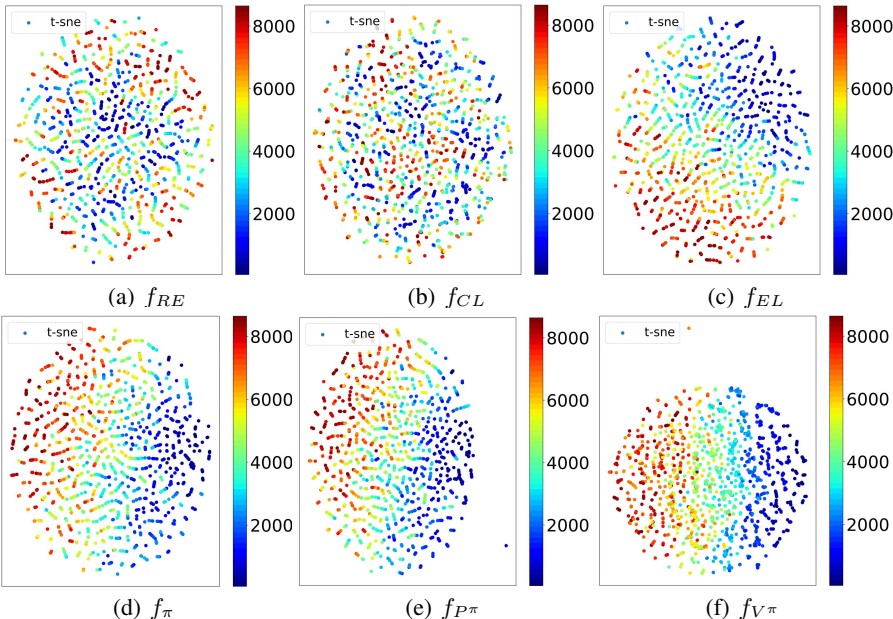

Figure 11: T-SNE visualization results of policy representation (Weak Generalization with 20% sampling ratio, IDP-v2). Each colored point represents a policy and the label of the color bar for all results is the expected return of the policy.

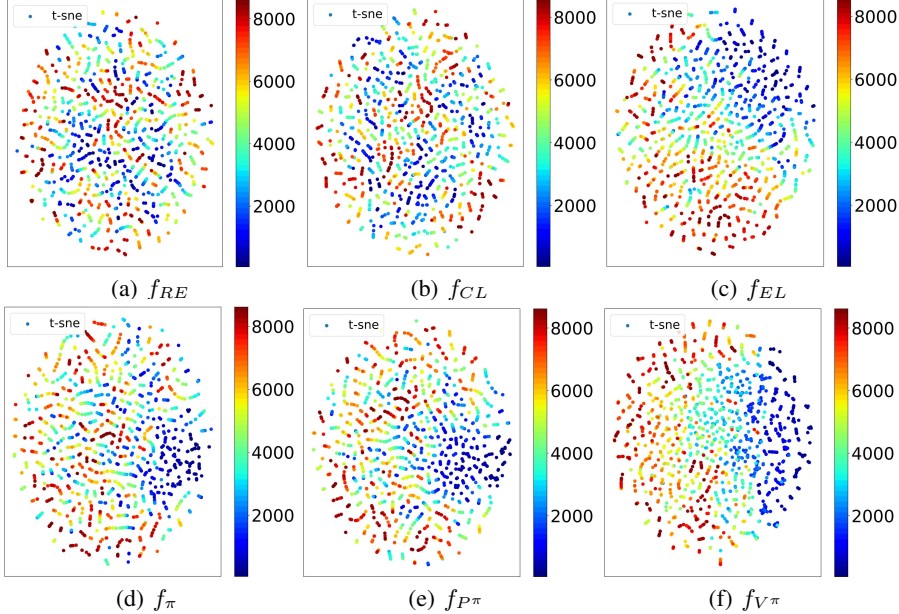

Figure 12: T-SNE visualization results of policy representation (Strong Generalization with 20% sampling ratio, IDP-v2). Each colored point represents a policy and the label of the color bar for all results is the expected return of the policy.

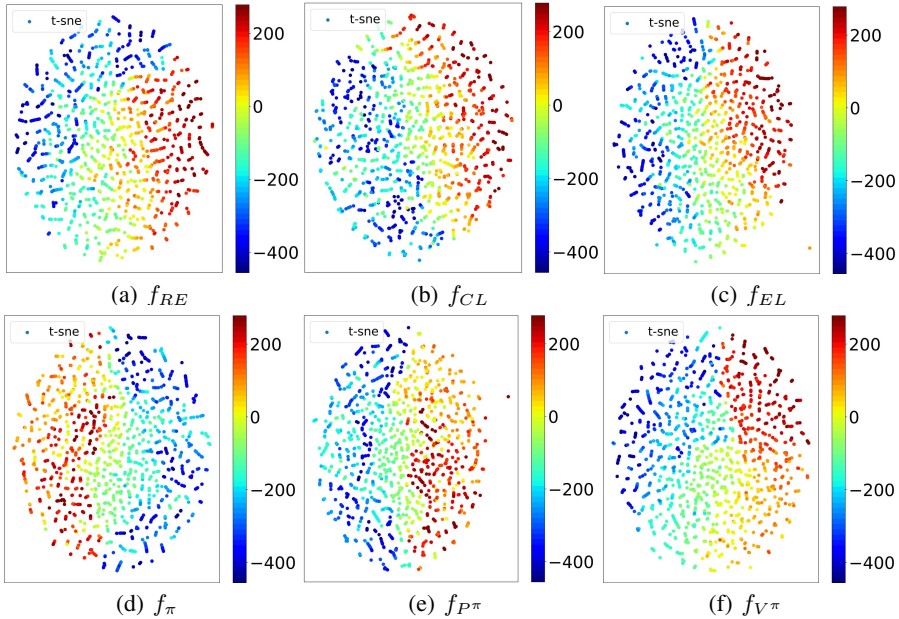

Figure 13: T-SNE visualization results of policy representation (Weak Generalization with 20% sampling ratio, LLC-v2). Each colored point represents a policy and the label of the color bar for all results is the expected return of the policy.

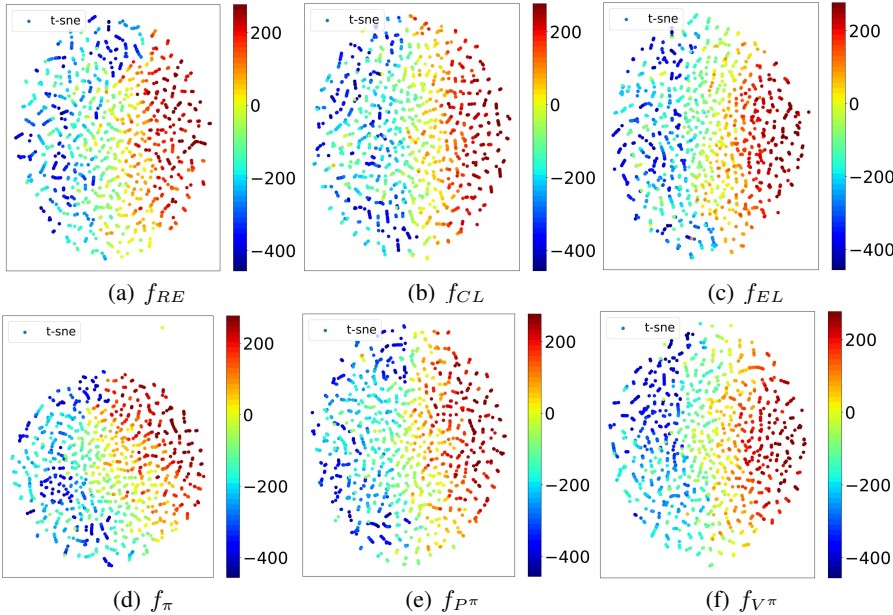

Figure 14: T-SNE visualization results of policy representation (Strong Generalization with 20% sampling ratio, LLC-v2). Each colored point represents a policy and the label of the color bar for all results is the expected return of the policy.

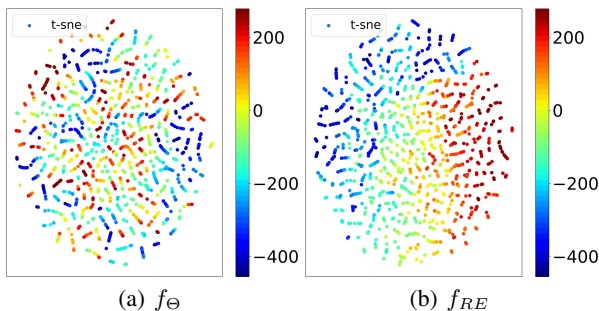

(a) $f_\Theta$          (b) $f_{RE}$

Figure 15: T-SNE visualization results of policy representation (Weak Generalization with 20% sampling ratio, LLC-v2). Each colored point represents a policy and the label of the color bar for all results is the expected return of the policy.

