# OpenReview forum: "Towards A Unified Policy Abstraction Theory and Representation Learning Approach in Markov Decision Processes"
_ICLR.cc/2023/Conference — Submitted to ICLR 2023_

### Official Review · Reviewer_M8fJ · 2022-10-20

**Confidence:** 4
**Correctness:** 4
**Technical Novelty And Significance:** 2
**Empirical Novelty And Significance:** 2
**Recommendation:** 5

**Clarity, Quality, Novelty And Reproducibility:**

This paper is well-written in general, but some important details are missing:
- How is $d_{P^\pi}$ and $d_{V^\pi}$ computed in the experiments? Does the algorithm know the transition model of the underlying MDP? How is the value function computed? To some extent, computing the value function of every given policy is not much easier than solving the optimal policy of the MDP.


**Strength And Weaknesses:**

### Strengths:

- The question of studied in this paper is interesting and relevant to many deep RL algorithms. For example, TRPO iteratively optimize its policy in a small neighborhood, whose definition requires a metric on the policy space. Different metrics induces different algorithms and it’s interesting to see how the choice of the metrics affects the performance.
- The results for value generalization on off-policy evaluation tasks are interesting, and raises several questions to ask in future works: does the same conclusion hold for more complex environments? Is it possible to use this method to off-policy optimization tasks? Does the method improve over other baselines?

### Weaknesses:

- The contribution of this paper is somewhat limited. The theoretical results are rather  straight-forward, and the empirical results are limited to grid-world environments (for policy optimization) and simple Mujoco tasks (off-policy evaluation). For the policy optimization experiments, the results only demonstrate that, unsurprisingly, the performance of TRPO and DGES depends on the choice of metrics. For off-policy evaluation, this paper only tests two tasks from the Mujoco environment.

- It is unclear to me whether a policy abstraction is necessary for online RL. Many algorithms for online policy optimization do not require a metric on the policy space, and whether TRPO with a good metric can outperform those algorithms remains a question.


**Summary Of The Paper:**

This paper studies the metrics (or abstractions) on the policy space for a given MDP. The paper defines three pseudo-metrics on the policy space: $d_\pi(\pi,\pi’)$ that measures the distance between the outputs of $\pi$ and $\pi’$ given a state, $d_{P^\pi}(\pi,\pi’)$ that measures the distance between the distributions of next state given a state and policy $\pi$ (and policy $\pi’$), and $d_{V^\pi}(\pi,\pi’)$ that measures the difference between the value functions of $\pi,\pi’$. This paper also provides an algorithm that learns a representation of policies such that the l2 distance of the representations is approximately equal to a given metric. Empirically, the metrics on the policy space can be used to improve algorithms such as TRPO and DGES on grid world environments, and can help value generalization on off-policy evaluation tasks.

**Summary Of The Review:**

My main concern is that the contribution of this paper is limited both theoretically and empirically. As a result, I recommend a rejection for the current version of this paper.

==== after rebuttal ====

I thank the authors for addressing my concerns regarding the relevance of policy abstraction in prior works. However, I am still not convinced that the contribution of this paper is significant enough. Therefore, I raised my score slightly, and encourage the authors to evaluate the policy abstraction algorithms systematically on benchmarking environments.

---

> ### Author Response · Authors · 2022-11-21
> **Initial Response to Reviewer M8fJ**
>
> We appreciate the reviewer's valuable comments. We try to address the concrete questions one by one in the following:
>
>
> **[Q1: How is $d_{P^\pi}$ and $d_{V^\pi}$ computed in the experiments? Does the algorithm know the transition model of the underlying MDP? How is the value function computed? To some extent, computing the value function of every given policy is not much easier than solving the optimal policy of the MDP.]**
>
> In this paper, we estimate the policy metrics (including $d_{P^\pi}$ and $d_{V^\pi}$) directly from the samples, bypassing estimating the empirical distributions (i.e., $\widetilde{\pi}$, $\widetilde{P}^{\pi}$, $\widetilde{Z}^\pi$) or requiring the access to the environment models.
> In particular, we adopt Maximum Mean Discrepancy (MMD) as the distribution metric (Section 4.2).
>
> **[Q2: It is unclear to me whether a policy abstraction is necessary for online RL. Many algorithms for online policy optimization do not require a metric on the policy space, and whether TRPO with a good metric can outperform those algorithms remains a question.]**
>
> In addition to TRPO and DGES, we consider there are some other online learning problems where policy abstraction (or representation) can be useful:
>
> 1) The OPE setting considered in our work is closely related to the policy evaluation training process in online RL. As in algorithms like PPO, TD3 and SAC, at any time point during the learning process, we have historical policies and experiences.
> One concrete example is PeVFA [1], which proposes learning policy representation from policy parameters or trajectories and leveraging value generalization among policies based on policy representation along the policy improvement path.
> Following such an idea, they propose a new Generalized Policy Iteration for more efficient online RL in a general form.
>
>
> 2) For Self-Imitation Learning (SIL) [2], historical good behaviors (actions, states) can be imitated by making use of a representation guidance.
>
> 3) For MARL [3], policy abstraction and representation are useful in learning with multiple teammates or opponents.
>
>
> In fact, some of the recent works (as we summarize in Table 4) have already used different policy abstraction and representation for different online learning problems. As we mentioned, both the theory on policy abstraction and unified methodology are less studied in the literature. Our work makes very first efforts to fill up the vacancy.
>
> Beyond using policy abstraction (or metric, representation) for all kinds of policy regularization (e.g., TRPO, DGES, SIL) and generalization among policies (e.g., PeVFA and opponent modeling in MARL [3]),
> another promising approach is to establish a surrogate policy space based on policy abstraction (or metric, representation), where policies (and the surrogate space) can be optimized (or co-optimized) in a more favorable manner.
> We consider this as a major direction of future work.
>
>
> &nbsp;
>
> Reference:
>
> [1] What about Inputting Policy in Value Function: Policy Representation and Policy-Extended Value Function Approximator. AAAI 2022
>
> [2] Self-Imitation Learning. ICML 2018
>
> [3] Learning Policy Representations in Multiagent Systems. ICML 2018

---

### Official Review · Reviewer_LtQT · 2022-10-24

**Confidence:** 3
**Correctness:** 3
**Technical Novelty And Significance:** 3
**Empirical Novelty And Significance:** 3
**Recommendation:** 6

**Clarity, Quality, Novelty And Reproducibility:**

This paper is well-written and easy to follow. Although the three policy metrics are not very new, the proposed unified theory of policy abstraction is novel and it provides a good tool for designing more practical methods of policy abstraction.

**Strength And Weaknesses:**

**Strengths**

1. The problem this paper considers is very important, and this paper provides a new method to solve it.
2. The literature review is sufficient. This paper has offered a detailed table of a taxonomy of prior policy abstractions under the policy abstraction theory in the Appendix.
3. This work is well-motivated. To solve the challenge of large scale and high complexity of policy space, policy abstraction representation might be a very promising way. And this work provides a unified theory for this line of works.
4. The empirical evaluation and theoretical results are of high quality. (1) Figure 1 is especially interesting and it clearly showcases the differences between the three policy metrics. (2) The definition of abstraction fineness is also interesting and might be a key property of policies.

**Weaknesses**

1. The empirical results are significant in some easy tasks and some specific algorithms. These results might be enough for this paper but the reviewer is still wondering whether there is a unified way to use policy abstraction to other algorithms that do not need policy metrics explicitly, such as PPO, for policy learning not policy evaluation. In other words, can $f_\psi$ in Equation 1 be directly used for policy learning?

2. This paper does not provide enough discussions of limitations.

**Summary Of The Paper:**

This paper focuses on providing a theory and methodology on policy abstraction and representation to reduce the high complexity of policy space in the Markov Decision Process. To achieve this, they first proposed a unified policy abstraction theory and discussed three types of policy abstraction. Then they generalize these policy abstractions to three policy metrics that can quantify the distance of policies instead of only binary signals. Further, they proposed a policy representation learning method based on the policy metrics. Empirically, they conducted experiments in both policy optimization and evaluation problems, which demonstrates the effectiveness of the proposed representation method.

**Summary Of The Review:**

This paper is a very complete work about policy abstraction and may inspire future related works. Although the optimality of the policy abstraction is not guaranteed or discussed, this paper still fill up the plank in both the theory and methodology of this sub-field.

---

> ### Author Response · Authors · 2022-11-21
> **Initial Response to Reviewer LtQT**
>
> We appreciate the reviewer's valuable comments and recognition sincerely, which provide helpful guidance to improve the quality of our paper.
>
> For the concrete questions and concerns, we try to address them one by one in the following:
>
>
> **[Q1: The reviewer is still wondering whether there is a unified way to use policy abstraction to other algorithms that do not need policy metrics explicitly, such as PPO, for policy learning not policy evaluation. In other words, can $f_\psi$ in Equation 1 be directly used for policy learning?]**
>
>
> We may take implicit policy metrics and using policy metrics (or representation) for policy learning rather than policy evaluation as two problems,
> although they can be coupled as in PPO pointed out by the reviewer.
> In addition, according to Lemma 2 (and 3) in [1], the clipping mechanism in PPO is equivalent to an explicit policy metric based on Total Variance.
> There may exist a niche for us to convert explicit policy metric into implicit forms or mechanisms.
> We appreciate the reviewer's inspiring comments.
>
> For other implicit policy metric, we consider it is possible to realize implicit policy metric by following in the principle of contrastive learning , non-contrastive learning (or mutual information optimization generally).
>
>
> For possible ways of using policy representation for policy learning (in addition to TRPO and DGES), we provide several angles below:
>
> 1) Policy metrics can be useful in Offline RL where a regularization on the deviation of learning policy from behavior policy (or a more advanced alternative).
>
> 2) With a policy representation space, it is promising to learn policy more favorably in the representation space.
> A few recent works make some preliminary efforts in this direction, e.g., learning to adapt the policy to solve new tasks in policy representation space [2,3], evolving and reinforcement learning in policy representation space [4].
>
> &nbsp;
>
>
> Reference:
>
> [1] Generalized Proximal Policy Optimization with Sample Reuse. NeurIPS 2021
>
> [2] Fast Adaptation to New Environments via Policy-Dynamics Value Functions. ICML 2020
>
> [3] PAnDR: Fast Adaptation to New Environments from Offline Experiences via Decoupling Policy and Environment Representations. IJCAI 2022
>
> [4] ERL-Re$^2$: Efficient Evolutionary Reinforcement Learning with Shared State Representation and Individual Policy Representation. arXiv:2210.17375
>
> **[Q2: This paper does not provide enough discussions of limitations.]**
>
> As we confess in our paper, our biggest limitation is the lack of theoretical results on the connection between policy abstraction and optimality in specific downstream problems. We think this is a fundamental and challenging problem and also the major direction of our future work.
>
> Another limitation, as pointed out by Reviewer DwZY, is the lack of discussion and investigation on state distribution in the definition of policy metric.
> We provide some discussions in **Q2 of the response to Reviewer DwZY**.
>
> For a more general one, explicit policy representation learning is more difficult than learning state/action representation due to the difference in their essential complexity.
> When considering a concrete learning process, the difficulty of policy representation learning can be escalated due to limited data (e.g., on-policy interaction samples) or complex form (e.g., parameters of policy neural networks).

---

> > ### Comment · Reviewer_LtQT · 2022-11-22
> > **Reply to the authors**
> >
> > Thanks for the detailed response. I'd like to maintain my score.

---

### Official Review · Reviewer_DwZy · 2022-10-25

**Confidence:** 4
**Correctness:** 2
**Technical Novelty And Significance:** 2
**Empirical Novelty And Significance:** 2
**Recommendation:** 3

**Clarity, Quality, Novelty And Reproducibility:**

- Clarity: while the writing is quite sloppy, interestingly, this does not take much away from clarity. However, at various points in the paper important results are pushed into the Appendix and their implications are omitted in the main text (see Weaknesses above), which does take away from clarity.
- Quality: I found the overall quality of the paper to be below average due to reasons listed under Weaknesses above.
- Novelty: I did not find the taxonomy of policy abstractions / similarity metrics proposed by the paper to be original or deeply insightful.

**Strength And Weaknesses:**

Weaknesses:

1. [Major] I think that the writing of this paper could be significantly improved and it needs copy editing (frequent grammatical mistakes, awkward wording, etc.).

2. [Major] Simply put, Thm. 3.1, which is the main theoretical result of the paper, is factually incorrect and therefore misleading as it stands. As discussed in the Appendix, as soon as the expected immediate reward depends on the action selection, $f_{P^{\pi}}(\pi_i) = f_{P^{\pi}}(\pi_j)$ does not imply $f_{V^{\pi}}(\pi_i) = f_{V^{\pi}}(\pi_j)$. The discussion in the Appendix downplays the importance of this case as a minority among interesting environments, but one can easily imagine, for instance, an Atari game (hardly a minority) where taking different actions in a given state yields different rewards despite transitioning the environment to the same next state. For a theorem to be mathematically correct, its surrounding conditions and assumptions must be listed explicitly. These cannot be deferred to the Appendix. The appeal of a simpler expression cannot take precedence over facts.

3. [Major] The formulations of the *-irrelevance abstractions and metrics are fairly trivial and should be readily apparent to a savvy RL audience, as are the coarseness relations among them that do hold true. For instance, it's quite obvious that $f_\pi \succeq f_{P_\pi}$ since $\pi_1(a|s) = \pi_2(a|s), \forall a \in A \Rightarrow P_{\pi_1}(s'|s) = P_{\pi_2}(s'|s)$.

4. [Major] I would have expected significantly more discussion on the choice of the state distribution $p$ over which the expectation of the distance is taken to compare policies. In Fig. 1, the doorway environment is used to dismiss $d_\pi$ as a metric that fails to show the difference, but as noted in the paper itself, this is only the case because a uniform $p$ is used. Then, isn't this a failure to choose the right $p$ for computing $d_\pi$ rather than a failure of $d_\pi$ itself?

5. [Major] It's not clear to me whether increased stochasticity in Fig. 1 indeed makes "for a better evaluation" and what "better" means. For example, in the case of $d_{V^\pi}$ the evaluation no longer makes sense as the value of both policies decay rapidly with increasing stochasticity since the probability of reaching the goal decays. Given both value functions $V^{\pi_1}$ and $V^{\pi_2}$ decay, it's not at all surprising that their difference would as well. Note that this can also be remedied by choosing $p$ to be concentrated around states adjacent to the goal.

6. [Major] The empirical evaluations on policy optimization only cover GridWorld environments. This would have been acceptable for a more theoretical paper, but the theory here is fairly simple so I would have expected evaluation on a larger number of more serious benchmarks (e.g., 3D continuous control). Furthermore, the shaded confidence areas in Fig. 2, especially 2a, show significant overlap between various approaches, which make it difficult to draw meaningful conclusions or be convinced of the claims made in the paper. Even if we take the mean curves, none of the policy metric learning approaches seem to significantly improve over the vanilla TRPO approach (which uses the maximum TV distance to measure policy distance). This result is once again deferred to the Appendix Fig. 4, but must be included in Fig. 2 in the main text in my opinion.

7. [Minor] Both, IDP and LLC are listed as MuJoCo environments, but I believe only IDP is a MuJoCo environment. The LunarLanderContinuous-v2 environment should be a simpler 2D environment from Brockman et al. (2016) to the best of my knowledge.

8. [Minor] It would have been nice if the paper engaged more with the literature on state abstractions and state similarity metrics.

9. [Minor] Near the end of Sec. 3.2, it is noted that a common choice for $D$ could be the KL divergence, but this is inconsistent with Def. 4, which requires that $D$ is a (pseudo-)metric. KL divergence is neither a metric nor a pseudo-metric, but a _divergence_.

Strengths:

- I appreciated the importance of studying policy abstractions and the attempt at providing a unified theory.
- I think the paper includes some interesting ideas and has a good set of initial experiments, which would make for a good publication after a major revision with better writing, more extensive evaluation and more precise claims & mathematical statements.

**Summary Of The Paper:**

The paper first highlights the observation that the policy space size can be prohibitive for policy search and off-policy evaluation when the  state-action space is large (e.g., high-dimensional continuous). Then, the paper aims to propose a unified theory of policy abstractions for determining equivalence between given pairs of policies analogously to the unifying state abstraction theory of Li et al. (2006). Three distinct policy abstractions are identified based on equality criteria for (i) action distributions, (ii) next-state distributions and (iii) expected value when the agent's behavior is conditioned on the respective policies. A theorem that attempts to establish a coarseness-based partial ordering between said abstractions (as well as two trivial extremal abstractions) is presented. The abstraction equivalence relations are relaxed to similarity metrics via an arbitrary statistical distance. Three GridWorld MDPs are considered to provide intuition on the differences between these similarity metrics. A policy representation learning approach is proposed: the approach minimizes an MSE error between policy embedding distance and an MMD-based realization of the target policy similarity distance. Empirical evaluations investigate the use of these metrics in the context of policy search on trust-region policy optimization (1st order) and diversity-guided evolutionary strategies (0th order) for GridWorld environments. Further empirical evaluations on Off-Policy Evaluation (OPE) performance are provided. These measure the ability to interpolate and extrapolate policy performance from policy parameters without interaction with the environment given prior policy-value pairs that are used to learn policy representations that respect policy distances in latent space.

**Summary Of The Review:**

My current recommendation for the paper is rejection. I am not convinced of the significance and validity of neither empirical nor theoretical results to a satisfactory extent, and further believe that a major revision is required to improve the writing and organization.

---

> ### Author Response · Authors · 2022-11-19
> **Initial Response to Reviewer DwZy(Part 1/2)**
>
> We appreciate the reviewer's careful and valuable comments very much.
> We would like the reviewer to know that the comments are of significance to the further improvement of this paper and our future work.
>
> **[Q1: Simply put, Thm. 3.1, which is the main theoretical result of the paper, is factually incorrect and therefore misleading as it stands. As discussed in the Appendix, as soon as the expected immediate reward depends on the action selection,  $f_{P^\pi}(\pi_i) = f_{P^\pi}(\pi_j)$does not imply $f_{V^\pi}(\pi_i) = f_{V^\pi}(\pi_j)$. The discussion in the Appendix downplays the importance of this case as a minority among interesting environments, but one can easily imagine, for instance, an Atari game (hardly a minority) where taking different actions in a given state yields different rewards despite transitioning the environment to the same next state. For a theorem to be mathematically correct, its surrounding conditions and assumptions must be listed explicitly. These cannot be deferred to the Appendix. The appeal of a simpler expression cannot take precedence over facts.]**
>
> We appreciate the reviewer for pointing out this. We have added the condition on reward function in Theorem 3.1 in the revised version for the correctness.
>
> As discussed in Appendix A.1, we divide the cases of reward function depending on state and action into two branches.
> We appreciate the reviewer for pointing out the Atari examples.
>
> For some more discussions,
> taking physical-world scenarios into consideration, we can hardly think of the environments where "taking different actions in a given state yields different rewards despite transitioning the environment to the same next state".
> As to the Atari examples, we would like to know if such cases are commonly seen **when we consider the "state" rather than "observation"**(since visual frames are partial observation in Atari).
>
>
> **[Q2: I would have expected significantly more discussion on the choice of the state distribution $p$ over which the expectation of the distance is taken to compare policies. In Fig. 1, the doorway environment is used to dismiss $d_\pi$ as a metric that fails to show the difference, but as noted in the paper itself, this is only the case because a uniform $p$ is used. Then, isn't this a failure to choose the right $p$ for computing $d_\pi$ rather than a failure of $d_\pi$ itself.]**
>
> We appreciate the reviewer's inspiring comments. We agree that the state distribution is a key point in the realization of policy metric (estimation).
>
> Let us consider some practical learning settings, we consider the major feasible state distribution is the on-policy state visitation distribution of the considered two policies (no matter approximating from on-policy samples or off-policy samples).
> Another possible general choice is to use a exploration policy to maximize the entropy of the distribution of visited states (approximating a uniform state distribution).
> Without extra knowledge, we may not use other possible distributions.
> We will take this point for further investigation in the future.
>
> As to the concrete case in Figure 1, we are examining the results when we follow the choice of state distribution we discussed above, to see how the metric curves would be different.
>
>
>
> **[Q3: It's not clear to me whether increased stochasticity in Fig. 1 indeed makes "for a better evaluation" and what "better" means. For example, in the case of $d_{V^\pi}$ the evaluation no longer makes sense as the value of both policies decay rapidly with increasing stochasticity since the probability of reaching the goal decays. Given both value functions $V^{\pi_1}$ and $V^{\pi_2}$ decay, it's not at all surprising that their difference would as well. Note that this can also be remedied by choosing $p$ to be concentrated around states adjacent to the goal.]**
>
>
> We follow the setting of Kanervisto et al. (2020) and increase the stochasticity.
> This purpose is to generalize the determinitic dynamics to the stochastic dynamics.
> This also provides some preliminary investigations on how different metrics are sensitive to the stochasticity.
> For example, the distribution-irrelevance metric is agnostic to environment stochasticity and the value irrelevance metric has poor robustness with the increase of environment stochasticity as the reviewer pointed out.

---

> > ### Author Response · Authors · 2022-11-19
> > **Initial Response to Reviewer DwZy(Part 2/2)**
> >
> > **[Q4: The empirical evaluations on policy optimization only cover GridWorld environments. This would have been acceptable for a more theoretical paper, but the theory here is fairly simple so I would have expected evaluation on a larger number of more serious benchmarks (e.g., 3D continuous control). Furthermore, the shaded confidence areas in Fig. 2, especially 2a, show significant overlap between various approaches, which make it difficult to draw meaningful conclusions or be convinced of the claims made in the paper. Even if we take the mean curves, none of the policy metric learning approaches seem to significantly improve over the vanilla TRPO approach (which uses the maximum TV distance to measure policy distance). This result is once again deferred to the Appendix Fig. 4, but must be included in Fig. 2 in the main text in my opinion.]**
> >
> > We appreciate the reviewer's valuable comments on experiments.
> > We will consider larger-scale environments for policy optimization in the future for useful empirical discoveries in practical problems.
> > Meanwhile, we will also consider to conduct experiments in demonstrative environments with fewer variables for more robust theoretical evidence, as suggested by other reviewers.
> >
> > **[Q5: Minors]**
> >
> > We thank the reviewer for pointing out these points. We have amended them accordingly in the revised version.

---

> > > ### Comment · Reviewer_DwZy · 2022-11-25
> > > **Thanks for your responses.**
> > >
> > > I have read the rebuttal and thank the authors for their responses. I would like to maintain my score at present.

---

> ### Comment · Area_Chair_DNij · 2022-11-24
> **Thank you! Are you satisfied by the answers?**
>
> Dear reviewer,
>
> Thanks again for your detailed review! The authors have replied back to you. Please read them carefully, and acknowledge their response. If there is still an unclear point about the paper or you do not agree with some of the responses, please let them know. We would like to have a robust discussion now.
>
> If you have any further questions from them, please ask them now. We have to make the final decision soon.
> Also as a courtesy to the authors, please acknowledge their rebuttal.
>
> Thank you,
> Area Chair

---

### Official Review · Reviewer_5gmd · 2022-11-02

**Confidence:** 4
**Correctness:** 3
**Technical Novelty And Significance:** 4
**Empirical Novelty And Significance:** 2
**Recommendation:** 5

**Clarity, Quality, Novelty And Reproducibility:**

The paper is clear and ideas are well structured. The contributions are novel, to the best of my knowledge. The empirical contributions are likely reproducible, with at most a few minor details omitted.

**Strength And Weaknesses:**

High-level comments
===============

The propose categorization of policy abstractions are interesting. Seeing as they are inspired by similar efforts for state abstraction and, more generally, by existing formalism of abstraction, I would expect these definitions to be provide useful categorizations future work can build off of. The accompanying theorem further strengths the theoretical contributions and my only complaint is that I which there was more.

My biggest issue is that many of the experiments don't seem to provide much insight on the properties of these abstractions. The empirical contributions are probably the weakest aspect of this paper. Overall, these contributions mostly involve retroactively interpreting and attributing meaning to the results of the experiment through the lens of the proposed abstractions, rather than focusing on more targeted experiments which provide insight on the benefits and limitations of these abstractions, e.g., with domains designed to highlight stark differences. When scaling to complex settings with approximations and many design decisions, it becomes difficult to know if some of the patterns we see are due to interpretation the author's provide or if it is just a result of the particular combination of choices and domain.

Detailed comments and questions
==============

Why is definition 4.3 based on distributions over returns while definition 3.3 is defined on expected returns, i.e., state values?

What discount factors are used? Notably, in Figure 1?

Sec 4.3 was hard to follow and took me a fair bit of effort/time to understand what the authors were doing. It feels like this can be explained a bit better. Maybe making equations more explicit or a diagram would help?

In addition to targeted experiments, experiments that vary fewer variables, e.g., small methods but different environment, can also provide some evidence if trends are robust. However, the empirical results found in Figure 2 vary both domain and the optimization method possibly conflating any trends.

Am I mistaken in saying that there is no original TRPO (which uses max KL-divergence as proxy for total variation if I recall correctly) as baseline?

Why does the performance of several of the TRPO variants decreasing?

Figure 2, why plot only 1/2 standard deviation?

Sec. 6, many of the comparisons seem to make strong conclusion from what seems like insignificant differences in results. In some instances, differences are well below the standard deviation. I don't think much can be said from these results, especially in the "strong generalization" case which uses only 5 seeds.


Minor nitpicks
===========

Paragraph 2, sec 4.2, should probably use Gretton et al. 2012 as main reference.

Eq. 3, might be worth mentioning that the biased estimate was used instead of the unbiased one.

last para, sec 4.3, "policy buffer" doesn't tell me where they came from. I would recommend saying something like "past observed policies" or abstract it away more explicitly, e.g., "policy samples from some given dataset of policies".

Figure 2 (and several other places), "std" what? Standard error or standard deviation? This should be stated fully.

Figure 2, caption probably should mention that the two subplots show results for different environments (i.e., shouldn't be directly compared).

Table 2, it might be worth discussing training error when commenting on generalization. Do cases with the best test errors also have the best train error? The current table makes this tedious to deduce. The test and train error could be reported instead, using highlighting to show better gap. Alternatively, just discussing the train error in the text would do.


**Summary Of The Paper:**

The authors propose a theoretical framework to categorize policy abstractions based on what properties are preserved. The main contribution is the proposal of irrelevance-based definitions based on policy action distribution, state transition distribution and state values. The authors then propose corresponding pseudo-metrics that permit for more approximate notions of irrelevance. The theoretical contributions conclude by showing that proposed categories of abstraction can be ordered using a concept of abstraction fineness. The authors show some illustrative empirical results in the case where the proposed metrics and their relevant distributions can be computed and compared exactly.

For the purpose of learning policy representations, the authors then propose a scalable approximate method for optimizing these metrics based on optimizing an alignment loss where the metrics are approximated using maximum mean discrepancy, a RKHS based approach. Empirical results are provided for a policy optimization task using TRPO and a diversity-guided evolution approach comparing the behavior of when simultaneously learning and constraining with, in the case of TRPO, the policy representation. The final empirical contributions compare the metrics and learned representation's ability to generalize state-values in the context of off-line policy evaluation.

**Summary Of The Review:**

The base idea of this work is very interesting but isn't carefully examined. The theoretical results plus the illustrative experiment might not be sufficient on it's own, as is. The other contributions, the approximate policy representation learning approach and accompanying experimental results, don't seem particularly insightful and, in some cases, significant. For this reason, I'm on the fence as to whether this is ready for publication but I'm open to discussion.

---

> ### Author Response · Authors · 2022-11-19
> **Initial Response to Reviewer 5gmd**
>
> We appreciate the reviewer's valuable comments.
> We would like the reviewer to know that the comments are of significance to the further improvement of this paper and our future work.
>
> For the concrete questions and concerns, we try to address them one by one in the following:
>
>
> **[Q1: Why is definition 4.3 based on distributions over returns while definition 3.3 is defined on expected returns, i.e., state values?]**
>
> We appreciate the reviewer for pointing out this.
> We view value expectation (i.e., expected return) and value distribution(i.e., return distribution) as the two variants in the family of value-irrelevance abstraction or metric (including $f_{J^{\pi}}$ presented in Definition 6 in the appendix).
> We present value(-expectation)-irrelevance abstraction in Definition 3 since value expectation is more conventional in formalization;
> while we use value distribution in Definition 4 for the convenience of our practical estimation with MMD (since we may need to approximate the value function if we estimate policy metric based on value expectation).
> We added the definition of value(-distribution)-irrelevance abstraction in Definition 6 for completeness.
>
> **[Q2: What discount factors are used? Notably, in Figure 1?]**
>
> In Figure 1, we set the discount factor to 1. As we clarified in Appendix C, we obtain the estimated value $v(\cdot)$ of each state by estimating 1 - _[expected number of step to goal when starting from a given state]_.
>
> **[Q3: Sec 4.3 was hard to follow and took me a fair bit of effort/time to understand what the authors were doing. It feels like this can be explained a bit better. Maybe making equations more explicit or a diagram would help?]**
> To provide a more intuitive explanation, we supplement an illustration for
> policy encoder (Layer-wise Permutation-invariant Encoder (LPE) in Appendix F.2 and Figure 5.
>
>
> **[Q4: Am I mistaken in saying that there is no original TRPO (which uses max KL-divergence as proxy for total variation if I recall correctly) as baseline? Why does the performance of several of the TRPO variants decreasing?]**
>
> The practical TRPO algorithm uses mean KL as a convenient (tractable) surrogate of max KL.
> Essentially, the trust-region optimization objective in original TRPO is equivalent to the one used in our TRPO-$f_\pi$ (shown in Figure 2(a)), with only difference in the metrics used, i.e., KL (original) v.s., MMD (ours).
> As shown in Figure 2(a), TRPO-$f_{{\pi}}$ outperforms the others. We consider that it is because $f_{{\pi}}$ follows the abstraction criterion regarding action distribution, thus pertains to the essence of TRPO.
>
>
> For the slight performance decrease after the TRPO variants reach their peaks in Figure 2, our methods reach peak quickly and then decrease slightly. we hypothesize that it may be due to the fact that in the current experiment, our trust-region threshold does not adapt to the entire optimization process.
>
> We appreciate the reviewer's comments, which motivate and inspire us to dive in a deeper investigation on these points.
>
> **[Q5: Other comments on our experiments]**
>
> We appreciate the reviewer's careful comments again. Your suggestions are very valuable for us to further improve our work. In the future, we may adjust our experimental perspective to provide more robust conclusions about our theory.
>
> **[Q6: Minors]**
>
> 1) [Re: Paragraph 2, sec 4.2, should probably use Gretton et al. 2012 as main reference.]
> Thank you for pointing out it. We have amended it in the revised version.
>
> 2) [Re: last para, sec 4.3, "policy buffer" doesn't tell me where they came from. I would recommend saying something like "past observed policies" or abstract it away more explicitly, e.g., "policy samples from some given dataset of policies"]
> In the revised revision, we have replaced "policy buffer" with "policy samples from some given set of policies".
> In general, such a set of policies can be an offline dataset or collected during an online learning process.
> 3) [Re: Figure 2 (and several other places), ”std” what? Standard error
> or standard deviation? This should be stated fully.]
> In our paper, "std" represents standard deviation. We have modified "std" to "standard deviation" in the revised version.
>
> 4) [Re: Figure 2, caption probably should mention that the two subplots show results for different environments (i.e., shouldn't be directly compared)]
> As suggested, we have clarified the environments corresponding to the two subplots in the
> caption of Figure 2.

---

> ### Comment · Area_Chair_DNij · 2022-11-24
> **Thank you! Are you satisfied by the answers?**
>
> Dear reviewer,
>
> Thanks again for your detailed review! The authors have replied back to you. Please read them carefully, and acknowledge their response. If there is still an unclear point about the paper or you do not agree with some of the responses, please let them know. We would like to have a robust discussion now.
>
> If you have any further questions from them, please ask them now. We have to make the final decision soon.
> Also as a courtesy to the authors, please acknowledge their rebuttal.
>
> Thank you,
> Area Chair

---

### Author Response · Authors · 2022-11-19
**Common Response**

We appreciate all the reviewers' valuable and inspiring comments.
The revised version have been uploaded. Individual responses will be uploaded soon.

For our revision, major updates are summarized below:

- We polish the paper and correct the grammatical errors and ambiguous contents of the paper.
- As pointed out by Reviewer DwZy, we amend Theorem 3.1 for correctness by adding an condition of reward function (which were discussed in the appendix).
- We supplement an illustration for policy encoder (Layer-wise Permutation-invariant Encoder (LPE) to provide a more intuitive explanation

As mentioned by the reviewers, our main theoretical contribution in this work is to propose a unified theory of policy abstractions and to provide a theoretical analysis of the properties for different policy abstractions.
Probably, the most important thing to be investigated in the future is the optimality of the policy abstractions in typical downstream learning problems.

For the experiments, we have to confess that the major obstacle that prevents us conducting more comprehensive experiments are the choice of downstream problems and the collection of policy data.
There exists no standard problem settings nor a standard protocol for policy data collection (e.g., dataset size, policy size, seeds, intervals). Moreover, the cost of evaluating each policy data point can expensive, e.g., 200 episodes rollouts.
This is also an important aspect we are working on to improve our experimental evaluation in the future.
Here we appreciate the reviewers' constructive comments and suggestions very much.

We hope our responses are helpful in addressing the questions and concerns raised.
**We are always willing to answer any of your concerns** about our work and we are looking forward to more valuable discussions.

---

### Decision · Program_Chairs · 2023-01-20

**Decision:**

Reject

**Justification For Why Not Higher Score:**

Three out of four reviewers believe that the paper isn't ready. There are several issues with the current paper. This is consistent with my own reading of the paper.

**Justification For Why Not Lower Score:**

N/A

**Metareview: Summary, Strengths And Weaknesses:**

The paper describes policy abstraction based on various notions of irrelevance: distribution-irrelevance, influence-irrelevance, and value-irrelevance. It shows the (topological) finenesss/coarseness relation between them. It suggests a policy representation learning method based on a similar concept and performs some empirical studies.

All reviewers believe that the categorization of policy abstraction is an interesting concept and the paper is addressing an important question. But most reviewers believe that more work is needed for this paper before it is ready for publication. The current theoretical result is rather simple and has room for more development. More work on the empirical side is also needed.

After reading the paper myself, I agree with their assessment. I encourage the authors to further develop this project, take the reviewers' comments into account in revising their paper, and submit to a future venue.